# Automated task training and longitudinal monitoring of mouse mesoscale cortical circuits using home cages

**Timothy H Murphy[1,2]\*, Nicholas J Michelson[1,2], Jamie D Boyd[1,2], Tony Fong[1,2], Luis A Bolanos[1,2], David Bierbrauer[1,2], Teri Siu[1,2], Matilde Balbi[1,2], Federico Bolanos[1,2], Matthieu Vanni[1,2†], Jeff M LeDue[1,2]**

[1]Department of Psychiatry, Kinsmen Laboratory of Neurological Research, Vancouver, Canada; [2]Djavad Mowafaghian Centre for Brain Health, University of British Columbia, Vancouver, Canada

**Abstract** We report improved automated open-source methodology for head-fixed mesoscale cortical imaging and/or behavioral training of home cage mice using Raspberry Pi-based hardware. Staged partial and probabilistic restraint allows mice to adjust to self-initiated headfixation over 3 weeks' time with ~50% participation rate. We support a cue-based behavioral licking task monitored by a capacitive touch-sensor water spout. While automatically head-fixed, we acquire spontaneous, movement-triggered, or licking task-evoked GCaMP6 cortical signals. An analysis pipeline marked both behavioral events, as well as analyzed brain fluorescence signals as they relate to spontaneous and/or task-evoked behavioral activity. Mice were trained to suppress licking and wait for cues that marked the delivery of water. Correct rewarded go-trials were associated with widespread activation of midline and lateral barrel cortex areas following a vibration cue and delayed frontal and lateral motor cortex activation. Cortical GCaMP signals predicted trial success and correlated strongly with trial-outcome dependent body movements.

**\*For correspondence:**
thmurphy@mail.ubc.ca

**Present address:** †University of Montreal, Montreal, Canada

**Competing interests:** The authors declare that no competing interests exist.

## Introduction

Variability in animal experiments can be reduced through automation of training and data acquisition procedures in combined behavioral assessment and brain imaging of headfixed mice. Automation has been extended to complex headfixed visual tasks with a relatively good success rate in mice (*Aoki et al., 2017*) and headfixed rats (*Scott et al., 2013*). While the work described in Aoki et al in mice was impressive for introducing a complex visual task to home cage training, this work did not autonomously gather longitudinal mesoscale brain imaging data, or use with animals in a group and relied on daily supervised 2-photon imaging. Such supervised imaging following home cage training of individual animals will permit high-resolution microscopy in the context of extended automated training (*Aoki et al., 2017*). Recently, our lab and others have begun to automate the acquisition of mesoscale functional brain imaging (*Murphy et al., 2016*) and motor actions data using compact and cost-effective Linux-based Raspberry Pi or other single board computers (*Ardesch et al., 2017*; *Silasi et al., 2018*; *Woodard et al., 2017*). This work extends previous automated analysis of animal behaviors within home cages (*Bollu et al., 2019*; *Robinson et al., 2013*; *Robinson and Riedel, 2014*; *Robinson et al., 2018*) to the more complex brain circuit level analysis during behavioral tasks in combination with mesoscale imaging. Such automated studies offer potential for 24/7 data acquisition from large numbers of animals in a relatively unperturbed manner. While automated mouse (*Murphy et al., 2016*) and rat brain imaging has recently been described (*Scott et al., 2013*), the work presented here provides a significant improvement over the initial mouse-based designs (*Murphy et al., 2016*) and to led to higher rates of head fixation, greater data throughput, and

coupling to a behavioral task. Furthermore, we provide extensive documentation on how to construct these cages and implement a head-fixing and task training protocol. Our refined cage allows the acquisition of datasets that co-register behavioral and functional brain imaging data, providing the basis for a flexible open-source tool for chronic experiments that could assess aspects of human neurological disorders modelled in mice.

## Results

### Improvements to the cage hardware

We describe results and procedures to build and use a new generation of home cages with emphasis on training for simple headfixed tasks during simultaneous autonomous acquisition of mesoscale brain imaging data (*Figure 1A,B* for cage scheme and *Figure 2* for training). We have tested 44 adult male GCaMP6 transgenic mice (*Table 1* and *Figure 3*) and an additional cohort of 8 female mice that were trained for ~7 h/day that yielded similar levels of participation (*Figure 3*, *Figure 3— figure supplement 1*). Mesoscale widefield in vivo imaging maps activation of cortex over a field-of-view that can extend bilaterally from olfactory bulb to visual cortex (*Mohajerani et al., 2013*). In contrast to single cell imaging needing high resolution in the X-Y plane and in depth, wide field imaging is typically done with larger focal volume (~2–3 mm) to image curved brain surfaces and lower spatial resolution (pixels binned to ~32–49 µm) to sum more photons over a larger area (*Lim et al., 2012*). Thus, the intrinsically large depth of field and reduced resolution provided by the small lenses employed by compact cameras such as the Raspberry Pi Camera Module (*Figure 1B,C*) are not expected to degrade signals. While we emphasize single photon mesoscale widefield imaging within these cages, our system would also be suitable for training animals on complex headfixed tasks where imaging could be performed using parallel manual headfixation and advanced two-photon imaging or electrophysiological procedures (*Chen et al., 2017*; *Le Merre et al., 2018*; *Prsa et al., 2017*), as was already elegantly applied by others (*Aoki et al., 2017*). We include an extensive supplemental guide with full construction illustrations, parts lists (see *Supplementary file 1* and *Table 2*), share files with Python acquisition and analysis code, and various CAD and electronics schematics in *Supplementary file 3* or as online resources.

The home cage platform captures wide-scale transitions in brain activity from windows of 8.2 × 8.2 mm with video rate temporal resolution (30 Hz) for several months using a double cage system (*Figure 1A*). Our original system for automatically head fixing mice, while effective (*Murphy et al., 2016*), did not result in daily durations of head fixation that would rival those in manually head-fixed mice for behavioral training or the integration of behavior with brain imaging of more complex tasks (which are in the order of 15–40 min sessions/day) (*Gilad et al., 2018*; *Guo et al., 2017*; *Guo et al., 2014*; *Prsa et al., 2017*). We have optimized the original mouse voluntary head fixing system to improve the number of head fixes per day and the total time head fixed for each cage (*Table 1*). To accomplish this, we have used several strategies: doubling the cage size (tandem cage) to include up to 8–10 animals and improving the mechanical fixation mechanism which made the process more palatable for mice (*Figure 1A,B*). The initial design (*Murphy et al., 2016*) employed electromagnetic solenoids. Using this system, it was impossible to control the degree of restraint and it produced a significant audible and vibration stimulus. As an improvement to this, we developed a servo-based system (Hitec HF-645 or HSB-9475SH; use the latter for even less noise and vibration) which allows variable degrees of headfixation with an initial training period where mice were loosely headfixed (achieved by controlling the servo travel) (*Figure 1B,C*). Loose headfixation allows mice to move significantly and grow accustomed to restraint. This loose fixation period, along with the less jarring mechanics has resulted in significantly higher rates of headfixation than previous work (*Murphy et al., 2016*). In a total of 5 improved cages that were tested, we observed average of 28 +−17 head fixations per day per animal for a subset of animals tested (23/44) that exhibited high rates of headfixation (for supplemental database of all mice see methods and repositories). The time headfixed for these 23 good performers averaged 18+−13 min/day per animal (*Table 1*, behavioral data from all mice are available in an online database, see methods). In the current cage the cut-off for a poor performer was <20 min of total headfixation time, while good performers averaged 15.8 +−19.9 hr (summed time over all daily sessions). A difference over previous work (*Murphy et al., 2016*) was removing mice from the cage that were poor performers, or insisted on entering the

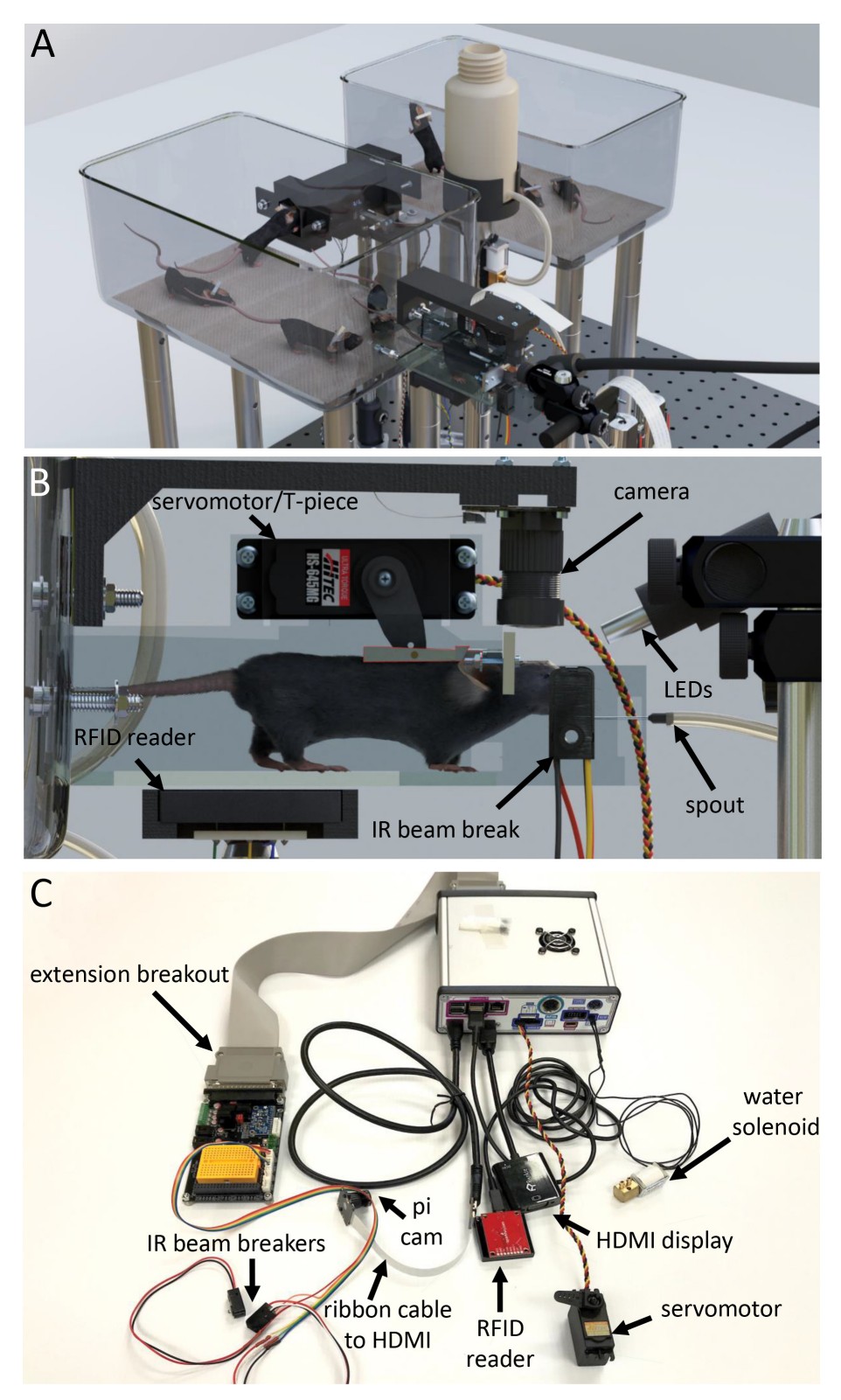

**Figure 1.** Tandem autoheadfixation and brain imaging home cage. (**A**) 3D graphic of the home cage system with 8 mice. Two cages are connected via a tunnel with an RFID sensor underneath and designed to automatically monitor the weight of the traversing mice. (**B**) Side view of the imaging chamber with a head-fixed mouse. An RFID sensor identifies the specific mouse. The snout of the mouse breaking the IR beam initiates the servo motor to head-fix the mouse by using the mounted aluminum T-piece to press the head-bar against the metal reinforced end of the chamber. Imaging LEDs,

*Figure 1 continued on next page*

Figure 1 continued
brain camera, and body cameras which record the behavior (eye, face and body) of the head-fixed animal are then activated. The spout provides the water rewards and detects licks for the task via a capacitance sensor. The Raspberry Pi processes inputs from the lick sensor and controls task related electronics (water solenoid, stimulus buzzer, auditory feedback). (C) Photo of the assembled major cage components. The Raspberry Pi and a printed circuit board (PCB) containing transistor logic switches and servo driving hardware are covered in a metal case (upper right). Electronic devices are plugged into headers on the PCB breakout (Water solenoid, Servo Motor) or into the Raspberry Pi directly (HDMI, USB RFID Reader). Picam with triple band filter 69013 m Chroma was used for brain imaging. IR beam breakers, auditory feedback buzzer, LEDs, vibration motor and licking sensor adapters (need short wiring to reduce noise) can be plugged via unique cables to the breakout extension board and to the Raspberry Pi. The breakout also supplies several regulated supply voltages and contains a small breadboard for use in developing future behavioral tasks.

headfixing tube sideways. Assuming a session length of ~40 s, which was the typical range in the previous system (*Murphy et al., 2016*), taking the top 50% of headfixing mice would result in an average of 6.5+−4 min of daily data acquisition or training per mouse using the initial design. In the improved cages, animals exhibit much longer daily durations of head fixation which leads to more spontaneous brain activity imaging data, but also the possibility to assess mesoscale functional circuits using GCaMP imaging as animal's train in simple detection or choice tasks. In total we were able to record 34,114 videos of brain activity in headfix sessions, where 29,097 were task related. An additional 5423 sessions were recorded from a cage with 8 female mice during daily lab-based 7 h/ day training where 5 advanced to high levels of participation (*Figure 3—figure supplement 1*). Moreover, the improved cage employs infrared beam breaks (*Figure 1C*) to determine mouse position for fixation rather than, as in the previous version (*Murphy et al., 2016*), electrical contact with the fixation bar where buildup of insulating dirt/oil can reduce sensitivity. A capacitance-based lick detector based around the MPR-121 that we employed previously in a mouse tapered beam assay (*Ardesch et al., 2017*) provides a system to confirm that mice are reaching the water spout and that they can register an output during a licking-based behavioral task. To relate the brain activity to the (currently) more than 7.5 million behavioral events (such as licking) and training circumstances we also implemented a relational database, allowing us to effectively analyze the data by pipelining plots and statistical tests to adaptive queries (see Materials and methods for how to access the database where we share all behavioral data collected).

## Initial training for headfixation and combined mesoscale imaging

For initial training 8–10 post-surgical male mice (2 cages of 10 and 3 cages of 8 mice) with RFID tags (*Bolaños et al., 2017*) and transcranial windows that were at least 45 days of age (*Silasi et al., 2016*) were kept in their double training cage for several weeks with ad libitum water from a bottle (*Figure 2A*, and see protocol steps in Methods). A group of female 8 mice was also trained in a similar manner (*Figure 3—figure supplement 1*). This ensures that the mice are well acclimated to the double cage (and each other) before headfixed training begins. Training begins by placing the mice within the headfixing cage and removing the water bottle overnight (*Figure 2A*). Mice were weighed on the subsequent day and this weight is the baseline by which 15% wt loss criteria were determined (for water supplementation). Mice received water from a spout for having their RFID tag read by a tag reader placed on the underside of the imaging chamber (see Materials and methods for details on spout positioning and strategies to prevent sideways entry).

After determining that the majority of mice entered the chamber (by reading RFID tags and lick detection) we progressively moved the water spout until it is 7–10 mm from the back wall of the headfixation tube (typically this takes 5–18 days) (*Figure 2A*). At this point mice are be able to break the infrared beams (*Figure 1B,C*) with their snouts and trigger blue LED lights used for mesoscale imaging producing what are termed no-fix trials. Once the infrared beams are broken a no-fix trial is initiated the mouse receives up to 5 large water rewards for just getting at the end of the tube. Once most of the remaining mice achieved this level (mice with very low entries were removed from the cage) we start the headfix training. If mice break the beam the servo now travels enough to prevent mice from leaving the tube, but the headbars can still freely move up to 20 mm along the rails of the tube (*Figure 2* and protocol steps Materials and methods). This step is termed loose headfixation. Initially we set a 50% probability of loose headfixation. Next, over 7–10 days we gradually tightened the fixation mechanism (increase servo forward travel, fixed setting) until the headbar was tightly pressed against the metal reinforcement at the end of the rails. Additionally, during this

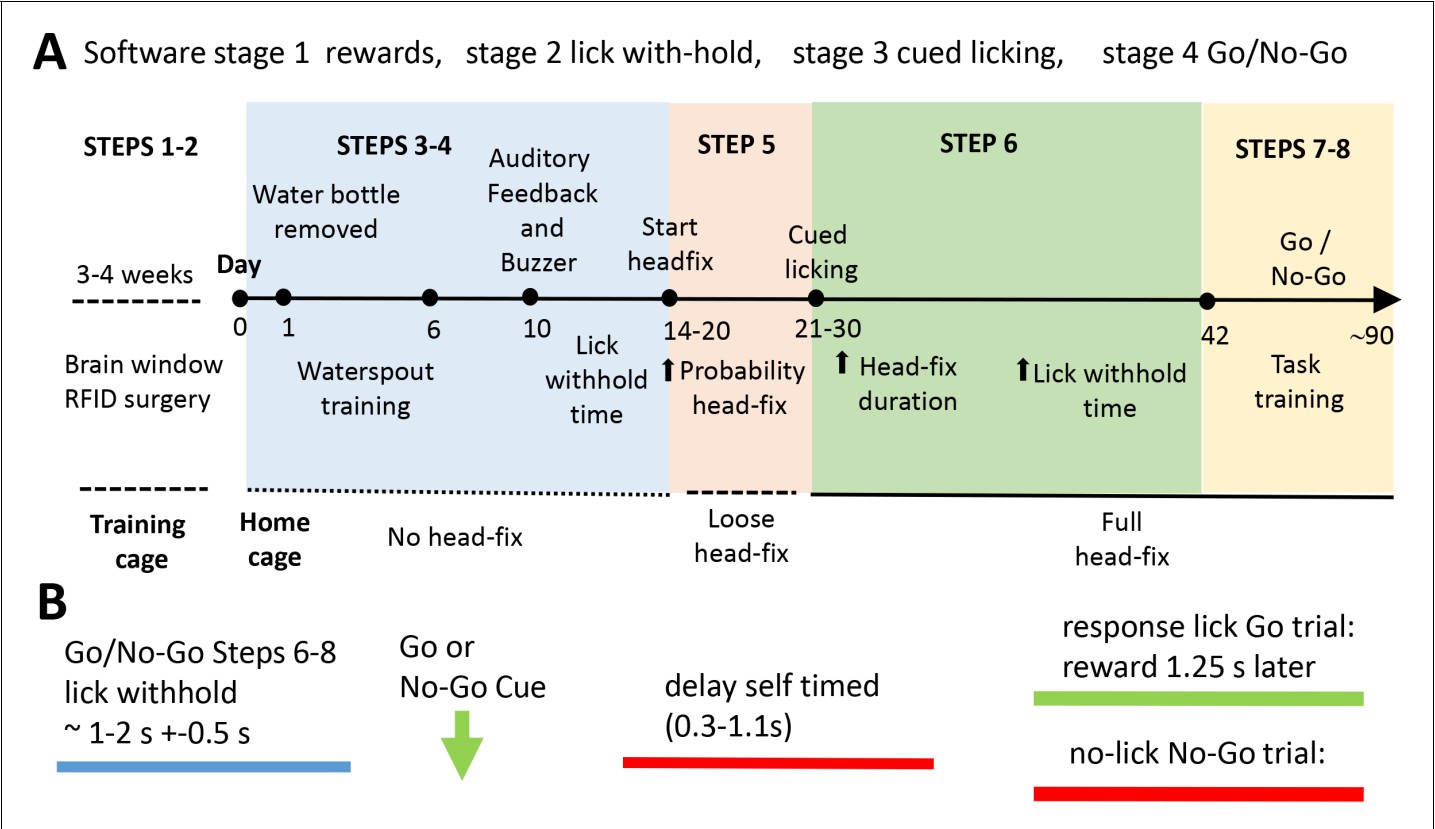

**Figure 2.** Scheme of the home cage training procedure leading to self-initiated head-fixing and task-based brain imaging. (**A**) Mice were kept in a training double-cage with a connecting tunnel to allow acclimation of pooled liters to achieve 8–10 mice of a particular genotype, sex, and age and receive ad lib water via a bottle (protocol step 1–2). After relocation to the home cage mice learned that water was provided from a spout near the entrance of the imaging chamber. Over time the spout is moved towards the end of the chamber (step 3). Once mice break the IR beam at the end of the chamber they were rewarded with more water. During this beam break training basic task trials can be introduced (step 4). Mice need to withhold licking for a short time to get a water drop after a buzz cue and receive auditory feedback for inappropriate licking. Requested lick withhold time can be increased each day. After mice learn the basic concept of lick withholding the servo motor was engaged and they are loose head-fixed for approximately 20 s with a random probability of 50%. The ratio of loose head-fixed trials was increased during the next week to ~70% and servo motor travel was increased further reducing head movement (step 5). When servo is tight enough to press the head-bar against the chamber metal stop the mice were trained to detect a stimulus (step 6). A licking response is now required in order to trigger water delivery. Additionally, the mouse has to delay its licking for some time after the stimulus (vibration) was given. During the next weeks head-fix duration and delay time are increased. In some cases no-go trials were introduced for cages with long runtime (steps 7–8). The acquisition software is organized by Python code stimulator objects that achieve specific broad training goals (these are termed stages: stage 1 rewards, stage 2 lick with-hold, stage 3 cued licking, stage 4 go and no-go). (**B**) General structure of a task trial within a headfixed session. To initiate a trial the mouse has to withhold its licking (blue) for an interval termed 'lick withhold time'. A vibration cue follows the cued licking (green arrow can be go or no-go based on buzz frequency). After the stimulus the mouse had to delay its licking for self-timed period (red) until the response licking period. Delay time was increased during step 6–8. After the delay time has passed the mouse needs to lick within 1.25 s in Go trials. If the mouse correctly responded to the go stimulus water is provided after the 1.25 s. In no-go trials the mouse has to further suppress licking waiting for the next stimulus. Auditory feedback was given for inappropriate licks in no-go trials. Delay time was first introduced in group 2. Groups 2–3 faced constant delay times (0.5 s and 0.25 s). Groups 4–5 experienced delay time training.

period we gradually increased the probability of headfixation until mice were fully headfixed 75–90% of the time for about 19–45 s. Headfixing probability was never set to 100% as a potential safety mechanism allowing animals that for some reason could not un-break the IR beams in time (parameter called skedaddle time in software) to be prevented from re-fixation. A text messaging service was also setup to send automated warnings if a mouse was present in the headfixing tube for longer than 600–800 s. Once mice were fully headfixed we continued using a single entrance reward as well as adding a licking task (*Figure 2*). Plotting entry rates, time headfixed, and headfixing rates for all mice tested indicated a peak of early entries just after the mice were placed in the cage (*Figure 3A*). At these early times of training (even in the second day in the cage) mice were actively licking the

**Table 1.** Headfixation statistics for 23 of 44 male and 5 of 8 female mice (lower panel) that were good performers.

6 of the 23 shown male mice were subsequently removed from the task due to health issues (2/6) or because they suddenly stopped performing (4/6). All together 23 of 44 mice were removed from the task, due to issues with the window and/or health (6/23), sideways entries (3/23) or poor participation (14/23) while 21 of 44 continued to perform. 4 mice of these latter 14 already stopped participating before headfixation protocol started. All behavioral data are available online in database where animal specific queries can be made (see Methods).Genotypes: Ai94-GCaMP6s (Group 1,3), Thy1-GCaMP6s (Group 2), TTA-GCaMP6s (Group 4,5, 6 females). Headfixed stats were calculated for the number of days with headfixation as indicated and only headfixed trials were used for calculation of task success rate.

| Male Mice Facility trained 24/7 | Group | Days under headfix protocol | Days with head-fixation | Total head-fixes | Total hours head-fixed | Head-fixes /day | Minutes head-fixed /day | Success rate GO trials of last 5 days [%] | Number of trials daily average based on 5 days at max success, all outcomes |
|---|---|---|---|---|---|---|---|---|---|
| 2016080026 | 1 | 90 | 60 | 933 | 9.8 | 15.6 | 9.8 | 91.3 | 17.6 |
| 201608466 | 1 | 93 | 91 | 2440 | 26 | 26.8 | 17.1 | 87.8 | 46.8 |
| 201608468 | 1 | 93 | 93 | 8360 | 97.7 | 89.9 | 63 | 8.6 | 1615.2 |
| 201608481 | 1 | 92 | 91 | 2197 | 24.2 | 24.1 | 15.9 | 92.8 | 222.6 |
| 201609136 | 1 | 90 | 82 | 1469 | 16.1 | 17.9 | 11.8 | 56.5 | 335 |
| 201609336 | 1 | 93 | 90 | 1344 | 11.9 | 14.9 | 8 | 70.4 | 23 |
| 210608298 | 1 | 92 | 88 | 2369 | 25.4 | 26.9 | 17.3 | 95.2 | 209.8 |
| 2016090793 | 2 | 43 | 32 | 1397 | 11.5 | 43.7 | 21.6 | 41.4 | 162.4 |
| 2016090943 | 2 | 47 | 37 | 1118 | 9.2 | 30.2 | 14.9 | 34.6 | 184.8 |
| 2016091112 | 2 | 13 | 7 | 141 | 1.1 | 20.1 | 9.1 | 6.1 | 28 |
| 2016090629 | 3 | 36 | 34 | 763 | 9.4 | 22.4 | 16.5 | 78.3 | 196.2 |
| 2016090797 | 3 | 33 | 30 | 866 | 10.8 | 28.9 | 21.7 | 75.8 | 173.2 |
| 2016090882 | 3 | 14 | 12 | 170 | 2 | 14.2 | 9.8 | 62.3 | 96.6 |
| 2016090964 | 3 | 9 | 8 | 144 | 1.4 | 18 | 10.6 | 58.8 | 100.5 |
| 2016090965 | 3 | 35 | 31 | 1024 | 12.8 | 33 | 24.8 | 68.8 | 185.8 |
| 2016090985 | 3 | 36 | 34 | 2072 | 25.9 | 60.9 | 45.7 | 0.4 | 581 |
| 2016091183 | 3 | 13 | 12 | 288 | 3.2 | 24 | 15.9 | 40.7 | 130.2 |
| 2016080252 | 4 | 85 | 73 | 1314 | 14.6 | 18 | 12 | 74.9 | 74.8 |
| 201608423 | 4 | 91 | 84 | 2513 | 29.9 | 29.9 | 21.4 | 73.1 | 359.8 |
| 201608474 | 4 | 35 | 33 | 632 | 6.7 | 19.2 | 12.1 | 42.9 | 199 |
| 801010205 | 5 | 22 | 21 | 368 | 2.8 | 17.5 | 8 | 31.5 | 92.8 |
| 801010219 | 5 | 36 | 34 | 803 | 7 | 23.6 | 12.4 | 34.6 | 153.4 |
| 801010270 | 5 | 34 | 27 | 362 | 3 | 13.4 | 6.7 | 30.2 | 130.4 |
| AVG | | 53.3 | 48 | 1438.6 | 15.8 | 27.5 | 17.7 | 54.7 | 231.3 |
| STDEV | | 32.4 | 31.0 | 1690.9 | 19.9 | 17.3 | 12.9 | 28.4 | 326.8 |
| SUM | | 1225 | 1104 | 33087 | 362.4 | 633.1 | 406.1 | | 5318.9 |
| Female Mice lab trained ~ 7 h/day | | | | | | | | | |
| 801010240 | 6 | 83 | 62 | 953 | 9.9 | 13.4 | 9.6 | 42.9 | 78 |
| 2018121234 | 6 | 83 | 39 | 812 | 8.8 | 14.5 | 13.6 | 43.5 | 93.4 |
| 2018121244 | 6 | 83 | 57 | 818 | 8.1 | 11.9 | 8.6 | 30.6 | 123.4 |
| 2018121245 | 6 | 83 | 68 | 1308 | 14.2 | 18.4 | 12.5 | 36.8 | 125.4 |
| 2018121379 | 6 | 83 | 70 | 1532 | 16.2 | 21.6 | 13.9 | 34.4 | 148.2 |
| AVG | | 83 | 59.2 | 1084.6 | 11.5 | 16.0 | 11.6 | 37.6 | 113.68 |
| STDEV | | 0 | 12.4 | 321.2 | 3.6 | 4.0 | 2.4 | 5.5 | 27.9 |
| SUM | | 415 | 355.2 | 5423 | 57.3 | 79.8 | 58.2 | | 568.4 |

waterspout (*Figure 3B*). This was followed by a transition to headfixation which began with a loose restraint and was progressively tightened over 4–7 days to the point where mesoscale imaging data could be acquired with minimal motion artifact (*Figure 3C*). The records of headfixation for all 44 male mice are displayed over a 45 day period (*Figure 3A–D*) (note, not all cages were examined for all 45 days and cage 3 was shut down due to a power outage during day 13–17). The graphs illustrated that the mice entered less frequently after the first 10 days of headfixation once they transitioned from entrance rewards to receiving multiple task rewards (*Figure 3A*) and collectively the mice of the 5 cages could yield up to 4 hr headfixed training or imaging a day (*Figure 3D*).

To quantify the observed clustering of headfixation events within and between mice we determined the time between headfixes for individual mice, as well as the time between a single headfixing mouse and any other mouse headfixing. Consistent with previous findings (*Murphy et al., 2016*) mice headfixed repeatedly over short time intervals (*Figure 4A,B*, red) with an average median interval of 5.8+- 4.1 s (*Figure 4C*, red line). Surprisingly, plotting histograms for the time interval between different mice headfixing (*Figure 4A,B*, black) also showed short interval repetitions with a second different mouse: average median interval of 305+- 575 s indicating linked behaviors between mice. Plotting the median separately for each group (*Figure 4C*, black line) showed large differences in inter-mouse headfixing times between cages, indicating potential group dynamics, however given

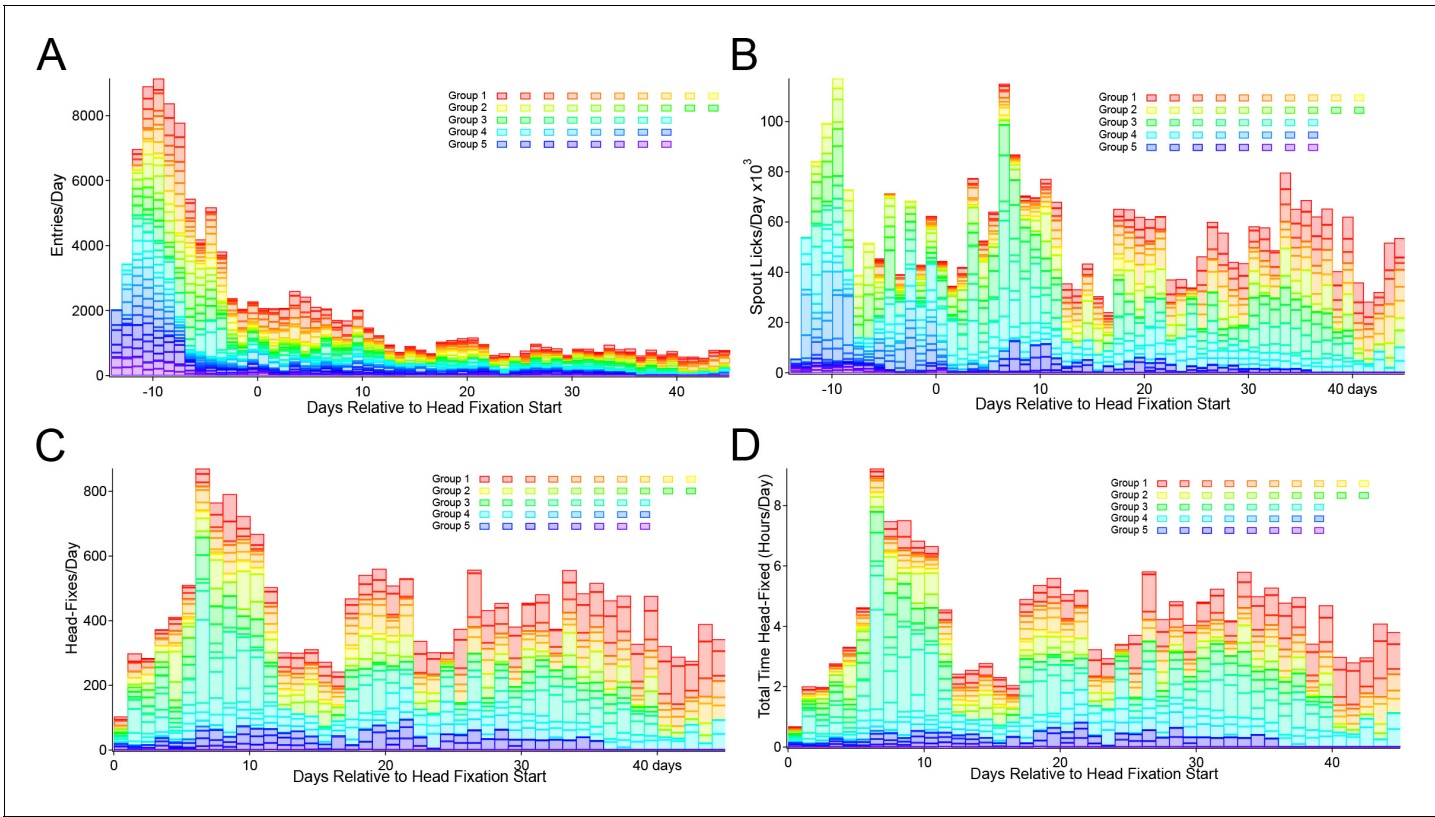

**Figure 3.** Entry, licking, and headfixing statistic for all mice tested. (**A**) Entries into the head-fix chamber, (**B**) spout licks, (**C**) head-fixes and (**D**) total time spent under head-fixation. All plots display participation relative to the starting day of loose head-fixing protocol (day 0). Entries are counted when the RFID sensor initially detects the presence of an RFID tag (implanted within the abdomen of a mouse). The lick detector in the first cage (group 1) was installed 6 days after beginning of the record, but before loose head-fixing protocol started. The 5 cage groups had slightly different runtimes and external interruptions of the systems (i.e. day 13–17 of Group 3 were not collected due to a power outage at the animal facility, animals were manually given water).

The online version of this article includes the following figure supplement(s) for figure 3:

**Figure supplement 1.** Headfixing and Go-trial statistics for 8 female mice training using ~7 h/day laboratory-based unsupervised training.

**Table 2.** Parts for Headfixing System.

| Description | # | Manufacturer | Part Number |
|---|---|---|---|
| Cage | | | |
| 1/4–20 Bolts, Setscrews, Nuts, Washers | | | |
| 8/32 Bolts, Nuts, Washers | | | |
| M2 and M3 Screws, Nuts, and Washers | | | |
| Cage 7.5' X 11.5' X 5' | 2 | Lab Products | 10027 |
| Aluminum Breadboard 18' x 24' x 1/2', 1/4'−20 Taps | 1 | Thorlabs | MB1824 |
| Ø1' Pillar Posts with 1/4'−20 Taps, 2' | 4 | Thorlabs | RS2 |
| Ø1' Pillar Posts with 1/4'−20 Taps, 3' | 8 | Thorlabs | RS3 |
| Ø1' Pillar Posts with 1/4'−20 Taps, 6' | 8 | Thorlabs | RS6 |
| Clamping Fork, 1.24' Counterbored Slot, Universal | 4 | Thorlabs | CF125 |
| Ø1/2' Pedestal Post Holder | 3 | Thorlabs | PH2E |
| Ø1/2' Optical Post, SS, 8–32 Setscrew, 1/4'−20 Tap, L = 8' | 3 | Thorlabs | TR8 |
| Ø1/2' Optical Post, SS, 8–32 Setscrew, 1/4'−20 Tap, L = 12' | 1 | Thorlabs | TR12 |
| Right-Angle Clamp for Ø1/2' Posts, 3/16' Hex | 2 | Thorlabs | RA90 |
| Ø25 mm Post Spacer, Thickness = 3 mm | 1 | Thorlabs | RS3M |
| Ø1.25' Studded Pedestal Base Adapter, 1/4'−20 Thread | 1 | Thorlabs | BE1 |
| Glass cut to 93 mm X 31 mm X 3 mm rectangle | 1 | Superglass | Custom |
| HS-645MG High Torque, Metal Gear Premium Sport Servo | 1 | Hitec | 32645S |
| HSB-9475SH brushless motor servo superior performance | 1 | Hitec | |
| RFID Reader ID-20LA (125 kHz) | 2 | IDInnovations | ID-20LA |
| RFID Reader Breakout | 2 | Sparkfun | 13030 |
| IR Break Beam Sensor - 5 mm LEDs | 1 | Adafruit | 2168 |
| NIR-Blocking Filters CALFLEX X | 1 | Qioptiq | G380227033 |
| RPi Camera (F), Supports Night Vision, Adjustable-Focus | 2 | Waveshare | 10299 |
| Flex Cable for Raspberry Pi Camera or Display - 2 meters | 2 | Adafruit | 2144 |
| Load Cell Amplifier - HX711 | 1 | Sparkfun | SEN-13879 |
| Micro Load Cell (0–100 g) - CZL639HD | 1 | Bonad/Alibaba | CZL639M |
| Capacitance sensor MRP121 | 1 | Adafruit | 1982 |
| Machined Parts (Stainless Steel) | | | |
| Head_fix_servo_coupler_PROFESH_revised_no_pinch.ipt | 1 | | Servo T-arm |
| ahf_contact_plate_L.stp | 1 | | Tube ending |
| ahf_contact_plate_R.stp | 1 | | Tube ending |
| 3D Printed Parts (Black PLA) | | | |
| Camera_Mount_V2.stl | 1 | | |
| head_bar_grabbing_plate_extended_6 mm_hole_in_bottom v25_reinforced.stl | 1 | | |
| Tunnel_Coupler.stl | 1 | | |
| Tunnel_Guider.stl | 2 | | |
| Tunnel_V2.stl | 1 | | |
| RFID_Holder.stl | 2 | | |
| Bottle_Holder.stl | 1 | | |
| AHF_HeadStraightener_25 mm.stl | 1 | | |
| AHF_HeadStraightener_50 mm.stl | 1 | | |
| 3D Printed Parts (Protolabs Watershed plastic) | | | |

*Table 2 continued on next page*

*Table 2 continued*

| Description | # | Manufacturer | Part Number |
|---|---|---|---|
| head_bar_grabbing_plate_extended_6 mm_hole_in_bottom V24 barrier1mmgreater_cut_front_bottom.stl | 1 | Protolabs special order | |
| Water reward parts valve, tubing etc. | | | |
| Ø1' Pillar Post, 1/4'—20 Taps, L = 12' | 1 | Thorlabs | RS12 |
| Water Solenoid | 1 | Gems Sensor | 45M6131 |
| Male Luer 1/16 | 2 | Component Supply Co. | LN-ML-062 |
| Polyurethane Tubing 1/8"ID X 3/16"OD | 1 | Component Supply Co. | PUT-02A |
| Med/Surgical Tubing 1/16"ID X 1/8"OD | 1 | Component Supply Co. | TND65-062A |
| 22 Gauge Needle 1.5 Inch | 1 | Becton Dickinson | Z192473 |
| 250 ml Water Bottle | 1 | Thermo Fisher Nalgene | 2003–0008 |
| 3D Printed Parts (Black PLA) | | | |
| Triple Light Guide and Imaging Parts | | | |
| Triple Bandpass Filter (camera) | 1 | Chroma | 69013 m |
| Liquid Light Guide | 1 | Thorlabs | LLG0338-4 |
| SM1 Adapter for Liquid Light Guide | 1 | Thorlabs | AD3LLG |
| SM1 Lens Tube, 3.00' Thread Depth | 3 | Thorlabs | SM1L30 |
| SM1 Lens Tube, 1.00' Thread Depth | 3 | Thorlabs | SM1L10 |
| SM1 Lens Tube, 2.00' Thread Depth | 1 | Thorlabs | SM1L20 |
| SM1 Retaining Rings | 2 | Thorlabs | SM1RR-P10 |
| Dichroic Cage Cube | 2 | Thorlabs | CM1-DC |
| Cage Cube Connector | 1 | Thorlabs | CM1-CC |
| Compact Clamp with Variable Height | 1 | Thorlabs | CL3 |
| Bi-Convex Lens | 4 | Thorlabs | LB1761 |
| AT455DC Size: 26 * 38 mm | 1 | Chroma | AT455DC |
| 25 mm x 36 mm Longpass Dichroic Mirror, 550 nm Cutoff | 1 | Thorlabs | DMLP550R |
| Ø1' Bandpass Filter, CWL = 620 ± 2 nm, FWHM = 10 ± 2 nm | 1 | Thorlabs | FB620-10 |
| ET480/30x Size: 25mmR R = Mounted in Ring | 1 | Chroma | ET480/30x |
| Ø1' Bandpass Filter, CWL = 440 ± 2 nm, FWHM = 10 ± 2 nm | 1 | Thorlabs | FB440-10 |
| Royal-Blue (448 nm) Rebel LED | 1 | Luxeon Star | SP-01-V4 |
| Blue (470 nm) Rebel LED | 1 | Luxeon Star | SP-01-B6 |
| Red-Orange (617 nm) Rebel LED | 1 | Luxeon Star | SP-01-E6 |
| Machined Parts (Stainless Steel) | | | |
| Milled as-1.50_2_v2.SLDPRT | 3 | | |
| Spacer_with_wire_hole_as-.500_v2.SLDPRT | 3 | | |
| LED_mount_as-1.50_v2.SLDPRT | 3 | | |
| 3D Printed Parts (Black PLA) | | | |
| TripleLEDLightGuide_Base.stl | 1 | | |
| Light_Guide_Mount_V2.stl | 1 | | |

the small sample size of cages (n = 5) we did not investigate statistically. Cluster analysis indicated most headfixes occurred in large temporal groups that could involve 1 or several mice (*Figure 4C,D, E,F*). Plotting headfix events on a 24 hr timescale with each day stacked on the Y-axis (*Figure 4D,F*) indicated that headfixes are not done randomly, but often form dense clusters of activity and 69

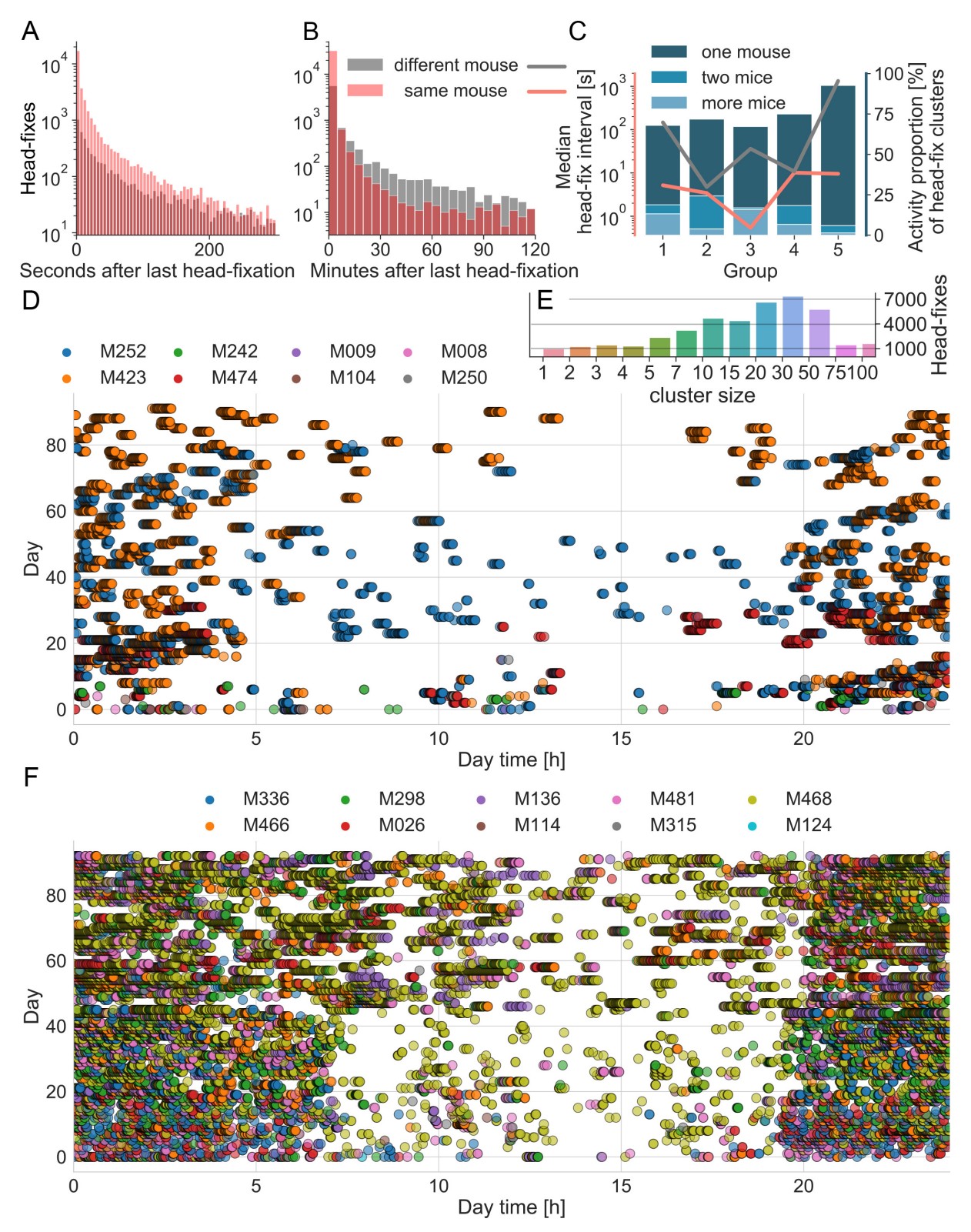

**Figure 4.** Group dynamics during headfixed behavior. (**A**) Histograms show the inter-headfixation interval for individual mice (same mouse) or for two different mice. Individual mice were most likely to start a new head-fix session within the first 10 s after the previous session. In about 90% of cases other mice were found to also enter within 60 s of an individual mouse (overlap darker red). Displayed bin size is 5 s. (**B**) Histogram plotted over longer time scale. After 10 min was more likely that the next head-fixation was done by another mouse than the same mouse as the gray area is greater

*Figure 4 continued on next page*

*Figure 4 continued*

(displayed bin size is 5 min) and after 2 hr the probability becomes equal again. Histograms include data from all 5 mouse groups during the time under head-fixing protocol. Group specific medians of the head-fix intervals can be found in C) using the same color code and the left (log) scale. (C) Distributions of different values over the 5 groups. Medians are displayed in red and black lines using the left (log) scale. The bar plots display the proportion of headfix clusters that include the activity of one, two or more mice, each bar sums up to 100%. (D) Group dynamics and diurnal variation in headfixation rates for cage group 4. Day 0 represents the beginning of the loose head-fixing protocol. Each circle represents a head-fix session and the colors correspond to unique RFID tags. (E) Count of head-fixes that were associated in clusters of distinct size resulting from KDE cluster analysis. Note that the displayed binning of cluster sizes was not linear and counts of larger cluster sizes were grouped together. (F) Data plotted for cage 1, note cage 1 started with 10 mice Ai-94 genotype.

+−11% of headfixes were performed at night (n = 5 cages); with the exception of cage 6 tested during the day only. Testing the time interval between headfixes revealed that they were not uniformly or normally distributed across a day (Anderson-Darling test; test statistic 14137, critical value for 1% level is 1.092, indicating a very low probability of normality) consistent with the temporal clusters we observed.

## Task training during combined cortical mesoscale imaging

Since headfixation was done in number of relatively short trials, mice were initially anticipating water delivery by vigorous licking following entrance rewards (*Figure 5A*). These high rates of licking can interfere with using licks within a behavioral task and will lead to brain activity largely reflecting frontal motor areas with the tongue (*Guo et al., 2014*). Therefore, the first stage of task training is called lick suppression where water is dispensed only when licking is withheld for 0.5–2 s. We added a 0.5 s mean random interval to this time to help ensure mice wait for a cue rather than use internal timing (*Figure 5A*); see group data on licking counts for all mice (*Figure 5B*). Licking at an inappropriate time was met with auditory feedback: a tone of 0.3 kHz was given to provide immediate feedback using piezo buzzer (Adafruit product #1739). Once lick suppression training was initiated mice were given a vibratory (Adafruit Vibe motor #1201) cue (500 ms continuous or pulsed $3 \times 200$ ms, 100 ms duty cycle) to signal that water will be dispensed after the lick suppression period is met. In cages 1 and 2 lick suppression began during headfixation, while in cages 3–5 it was started 2–4 days prior to loose headfixation. Over time the period of lick suppression was increased to 2 s (+- 0.5 s random interval) and water would only be dispensed when a mouse licks within a time window after the cue. As the goal was to extend training to a go and no-go task (*Gilad et al., 2018*), where mice will need to process and discriminate between two similar vibratory cues, we also added a self-timed delay period after the cue (0.3–1.1 s). Mice should not lick between the cue and the end of the delay period. When using the delay period, water was delivered following go-stimuli if the licking response occurred between the end of the delay period and 1.25 s after that (*Figure 5A*). For example, with a 1 s delay a mouse must lick between 1 and 2.25 s after the cue to receive a reward which comes 2.25 s from the cue. Initially, mice licked immediately after the cue and these errors were not rewarded and these task outcomes were termed lick early (coded −4) trials and signaled by auditory feedback using the same tone as for lick suppression (*Figure 5B*). Over time the delay period is increased from 300 ms by ~50–100 ms a day to an eventual 1100 ms. While we have termed these go and no-go trials, the addition of delay period (which in trained mice can be up to 1100 ms) makes the primary response after the vibratory conditioned stimulus (CS) a withholding of licking until after the delay period. Using this design all trials could be considered no-go and they would occur in the context of discriminating rewarded (CS+) and non-rewarded vibratory conditioning stimuli (CS-) that vary in frequency.

## Go task headfixed and non-headfixed task performance

Typically, we did not set the probability of head fixing to be 100%. In most cages headfixing probability was initially set low 25–50% and then moved to 70–90%. Having the head fixed probability lower than 100% was thought to aid to some mice which preferred to do the task in non-head fixed manner. Surprisingly, analysis of success rate data for the Go detection task indicated that most mice actually had higher success rates while head-fixed than non-head fixed (RM 2WAY ANOVA, p=0.0026, n = 8) (*Figure 5C*). All data on task performance presented in *Table 1* and used to calculate reported d′ were calculated for fully headfixed trials. We have not investigated the nature of this

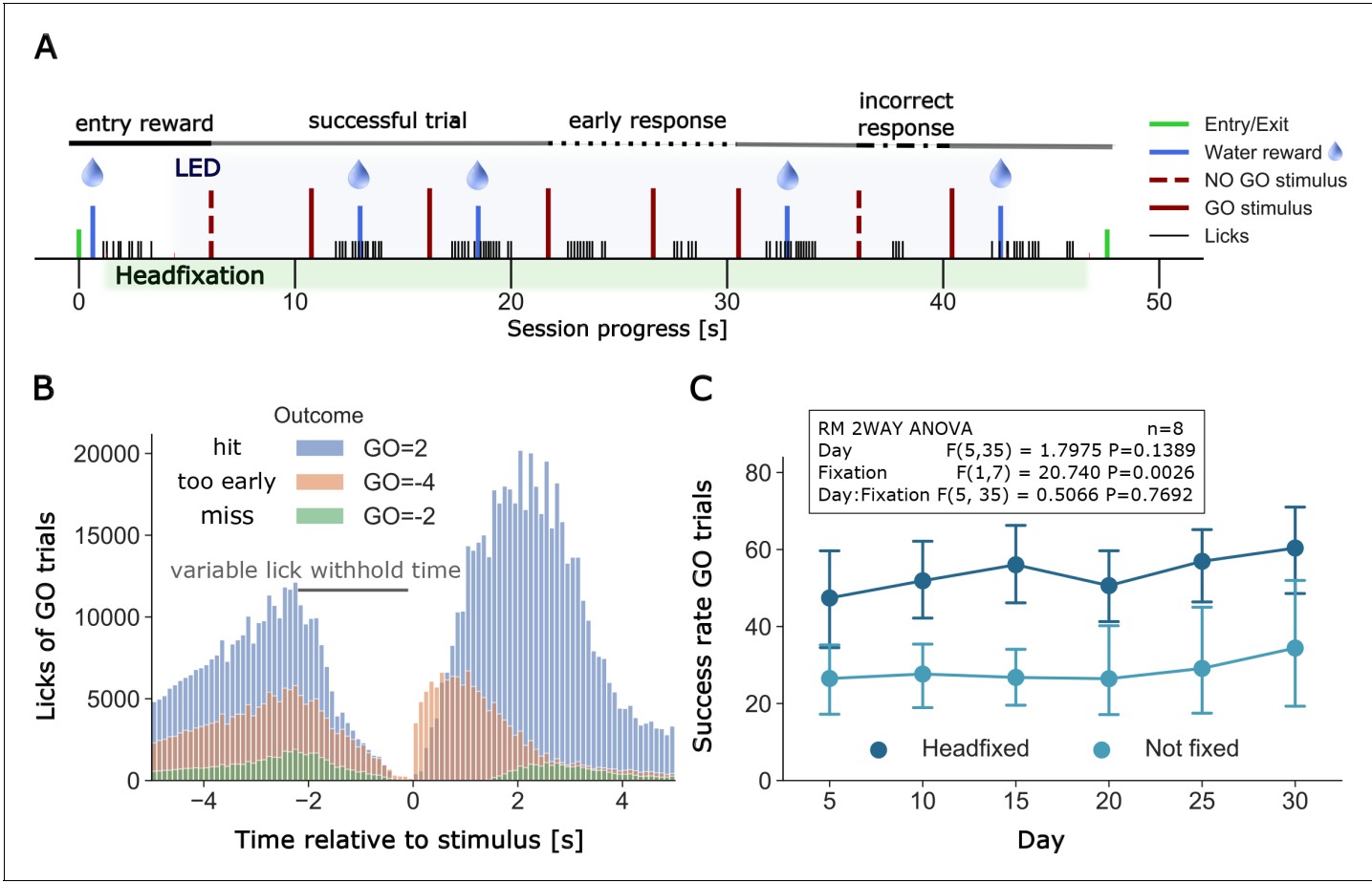

**Figure 5.** Example of a full headfixation session and group licking behavior during go-trial detection task. (**A**) The entering TTA-GCaMP6s mouse (RFID tag ID 201608423, Group 4, day 49 of head-fixation) was rewarded with a water drop and was head-fixed when it broke the infrared beam and brain and body behavioral video recording was started. The blue light LED directed to the brain window for GCaMP epi-fluorescence was turned on 3 s after the beginning of head-fixation and was switched off 3 s before the servomotor releases the mouse. Trial outcomes included in the displayed session scheme (in order left to right): GO = 1, GO = 2, GO = 2, GO=-4, GO=-4, GO = 2, GO=-1, GO = 2 (outcome shortcuts specified in *Figure 6B*). (**B**) Time distribution of licks of different outcomes of GO trials under head-fixation, note the delay period increases as training progresses. Plot displays all licks from 20 mice of groups 2–5. Mice of group 1 did not have a self-timed lick delay time and thus no GO=-4 outcomes and were excluded from the plot. Y-axis displays the raw count of licks (binned in intervals of 0.1 s) between −5 and 5 s after stimulus. Licks between two stimuli are displayed twice, considered as a lick before and after a stimulus (multiple stimulus trials up to 7 are given in a single headfixed session). Number of trials and licks due to outcome (trials/displayed licks/unique licks): GO=-2 (6181/65225/58538), GO=-4 (14501/295624/263467), GO = 2 (30005/823637/661619). Outcome shortcuts: (GO=) 2: successful go trial, −2: unsuccessful go trial (no/late response), −4: unsuccessful go trial (early response, licked within delay time). (**C**) Repeated measurements two-way ANOVA comparing success rates of go trials between head-fixed and not head-fixed trials. Mice of group 2–5 had a random chance (30%) of getting a session without head-fixation. Procedure and task were the same as in head-fixed sessions. Success rates of mice that performed over a period of 30 days (n = 8) were binned in 5 day intervals. The ANOVA shows a significant lower success rate of non-fixed go trials.

difference further, although it could suggest that head fixation helps to constrain mouse behaviors similar to that observed in reaching tasks that were not impaired by headfixation (*Galiñanes et al., 2018*). While the anecdotes of better task performance in headfixed mice are interesting, another explanation is that because there were more headfixed trials than in the non-headfixed state and animals were simply less practiced and not fully engaged during these trials.

Having longitudinal data also allowed us to assess whether mouse performance differed in the day or night period. In each of the 21 mice that performed the go task for more than 10 days we assessed whether they had a consistent diurnal preference for the licking task resulting in a significant difference in success rate. Overall, the ratio of success rate for day and night was similar across

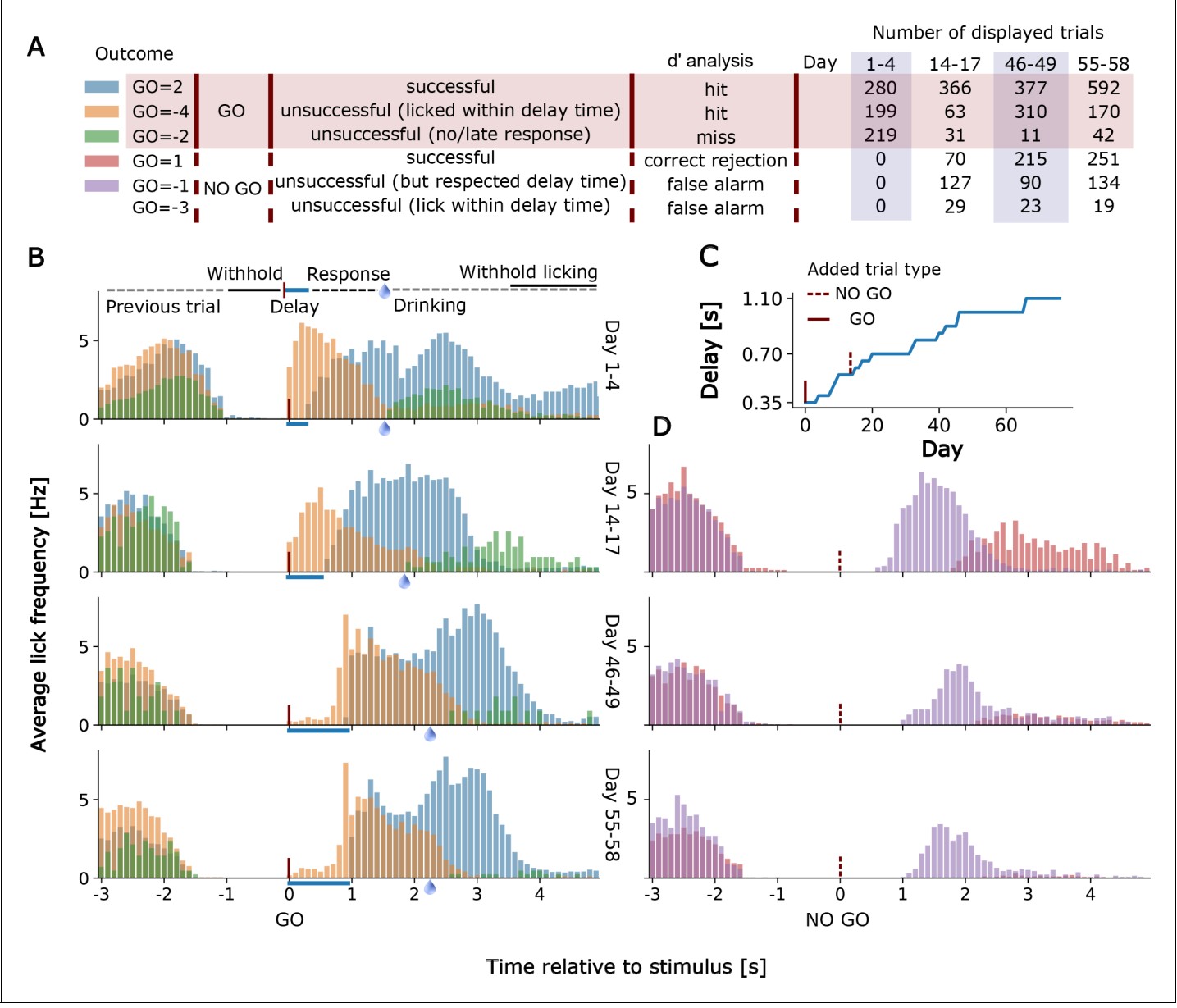

**Figure 6.** Go and No-Go Task training for an individual mouse. (**A**) Shows the number of trials performed, outcome software shortcuts and their consideration in the d' analysis of displayed trials for TTA-GCaMP6s mouse RFID tag ID 201608423. Go=-3 trials are not displayed in D) due to their rare occurrence. (**B**) Lick histograms for go trials on different training days. If the mouse withholds its licking for a given amount of time (up to 2 +- 0.5 s) a new trial is initiated by a randomly given go (or no-go) cue. Correct no-go trials are unrewarded, correct go trials are rewarded if the mouse delays its licking response for a given delay time. After the delay time has passed the mouse has 1.25 s to respond with licking. A water reward is given when the 1.25 s have passed and the mouse responded within this 1.25 s time window. A new cue is given when the mouse withholds its licking again for approximately 2 s. Blue bars show the lengthening of the delay time. Water drops indicate when the reward is given. (**C**) Delay times used and (**D**) No-go trial lick histograms.

mice (1.03+0.17, t-test: p=0.40, maximum ratios: 0.56 and 1.39, n = 21 mice that performed the go-task under headfixation for more than 10 days). We also assessed whether any individual mouse had a consistent day-night preference (t-test for each mouse using day/night success ratios for each day) and found no instances of a significant bias (p>0.05), except M985 that had a very low success rate (see *Table 1*) and therefore produced a very inconsistent daily day and night success rate ratio.

## Lick detection and training of a single mouse over an early, intermediate, and advanced days of in cage training

In *Figure 6* we show examples of lick detection and training of a single mouse over an early, intermediate, and advanced day of in cage training. The results of training are scored automatically for 6 possible task outcomes (*Figure 6A*). Mice were first trained on the go detection task and over the first 1–4 days could already pause and lick correctly (Go=+2 code) after the delay period. Even early in training there was relatively little non-rewarded early licking (Go=-4 code for early licking in the go trial (*Figure 6B*). The proportion of early licks (Go=-4) and ignored cues (Go=-2) was further reduced after 55–68 days of training even with a much longer delay period of up to 1 s (*Figure 6C*). No-go trials were introduced after 14 days of go-training and initially mice had trouble distinguishing the 2 trial types and would lick after the delay period as in a go-trial leading to a −1 code error (or −3 when licking early during the delay period for a no-go trial). Throughout training, there was marked reduction in licking after the no-go cue as the mice learned to distinguish the two stimuli (No-go trial correct rejection is No-go=+1) (*Figure 6*).

While we appreciate the power of a choice task, the sample sizes for the no-go task were relatively small (n = 10 mice) and only 5 of these mice had a sensitivity index (also called d') greater than 2 during their best task training days. d' is a z-transformed measure of the discrimination between go and no-go stimuli (hit rate (correct)z – false alarm (incorrect)z), with a value of 0 corresponding to no discrimination. In the d' analysis we disregarded failures due to the delay time early lick errors (*Figure 6A*) to calculate the hit rate and false alarm rate as these nonetheless reflected stimulus detection. The 5 mice that were good performers had a d' that averaged 1.37+−0.53 over their days of testing and they reached a maximal daily d' that averaged 2.3 +−0.52 and ranged from 1.9 to 3.21 (M252, M423, M026, M098, and M336). The statistic itself can be used as an estimate of how far values are from the hypothetical mean or 0, with 1.37 and 2.3 being in the 92% and 97% tails of the distribution. The other 5 mice performing the go and no-go task had lower average d'=0.2 +−0.54 and only reached a mean d' of 0.98+−1.01 in maximal daily training values.

## Mesoscale cortical functional imaging within home cages

Mesoscale GCaMP signals were collected from the dorsal cortex every time the mouse headfixed and examples are shown in *Figure 7A*. While the focus of this manuscript is on the construction of cages and the training of mice, we present cortical GCaMP6 imaging examples to illustrate that high-throughput automated mesoscale imaging is possible and can be integrated with behavior. The name of each brain imaging file which contains both the mouse ID and the time stamp can be found in the SQL database (RFIDtag_xxxx_timestamp.raw; see Materials and methods for URL and hosted as a full text file archive on Zenodo (https://doi.org/10.5281/zenodo.3268838) for the 5 cages of male mice and cage 6 female mice https://doi.org/10.5683/SP2/9RFXRP. GCaMP cortical signals allow functional connectivity to be assessed using correlational methods such as seed-pixel analysis of the hindlimb somatosensory cortex (*Figure 7B*). The location of hindlimb cortex was estimated by overlaying an averaged cortical anatomical parcellation based on the Allen Institute for Brain Science mouse connectivity atlas and the approximate location of the bregma landmark (*Vanni et al., 2017*). In addition to the brain video showing normalized cortical functional GCaMP6 signals, up to 3 other video cameras were used to obtain behavioral data under infrared illumination (*Figure 7C,D,E*). Circuity was added to cages 3–5 to turn-off infrared beam breaks once the mouse was headfixed as the infrared beam-breaks interfere with the behavioral cameras. Behavioral videos and GCaMP signals were synchronized using a master TTL switch that controls the infrared and blue LEDs used for GCaMP excitation or hemodynamic correction. As a means of illustrating the potential relationship between moving body parts and cortical mesoscale networks we made maps of the correlation between brain GCaMP signal pixels and normalized behavioral video ROIs for tongue, whiskers, and hindlimbs with a lag time of zero (see example *Figure 7F*). This analysis indicated a network of areas around limb and facial cortices that extended into frontal motor areas associated with licking (*Guo et al., 2014*). Plotting movement from the video analysis ROIs alongside licking rates (from the capacitance sensor) and mesoscale GCaMP signal for anterolateral motor cortex indicated that trial-associated licking bouts were accompanied by large scale body movements involving oral-facial, hindlimbs, and whiskers and GCaMP signals on the level of individual trials (*Figure 7G,H*), for group data analysis see Figure 9.

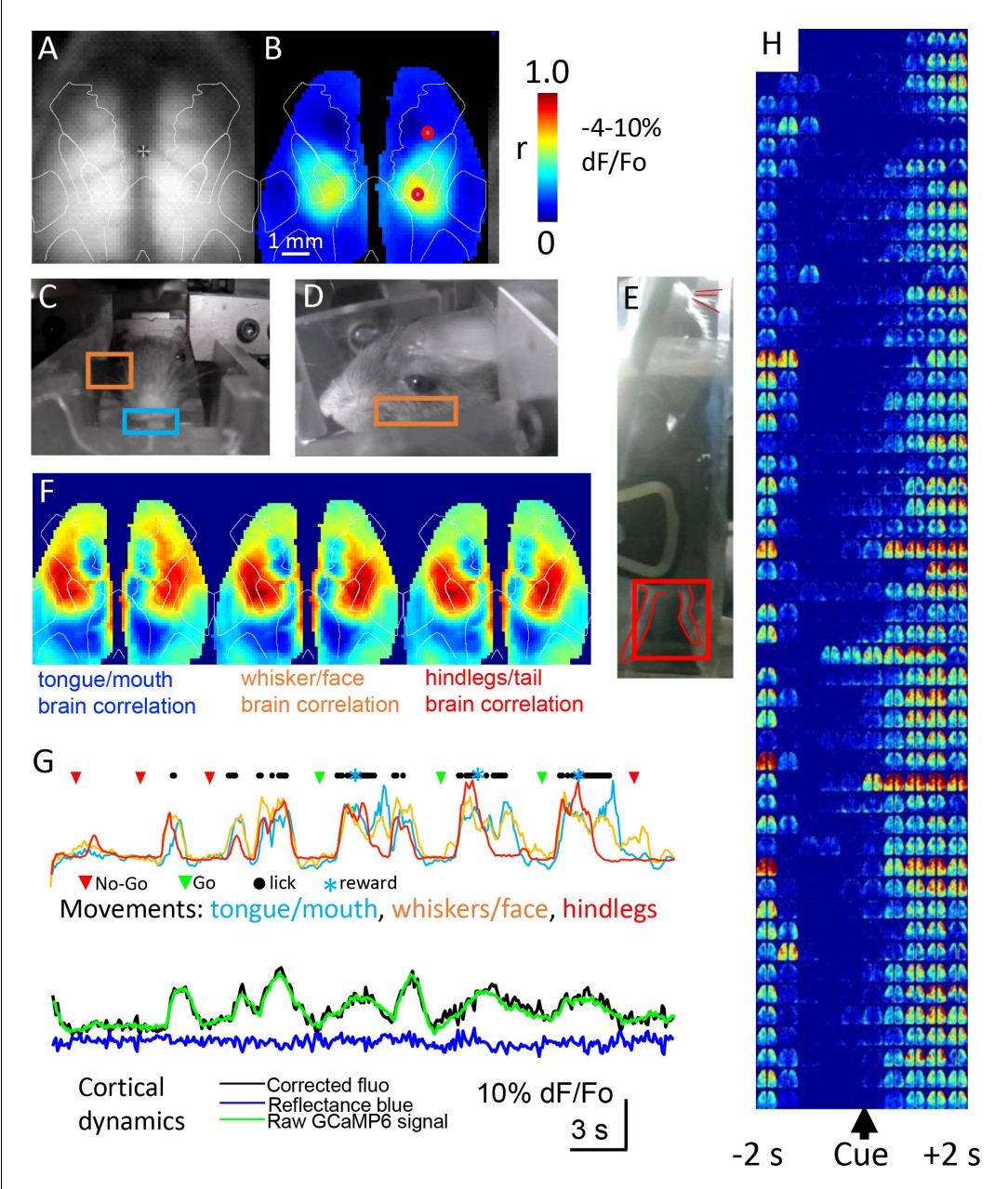

**Figure 7.** Mesoscale imaging of task-dependent GCaMP6 activity in automated home cages. (**A**) Raw fluorescence image of GCaMP and transcranial window, overlay represents a re-drawing from the *Oh et al., 2014*, note we have down-sampled the original data (256 × 256 pixels) to 64 × 64 pixels to speed processing. (**B**) Seed-pixel correlation map with seeded region in the right hind limb somatosensory cortex for mouse M8423 for data from 09252018 to 09292018 over which 151 headfixed sessions occurred (total imaging time of ~88 min, concatenated headfixed trials), red circle on the right hemisphere in the area of high correlation was the presumed hind limb area, the second red circle located more anterior was used for plotting ROI values in panel G. (**C, D, E**) Examples of behavioral ROIs for mouse tongue face, whisker, and hind leg, regions of interest (ROIs colored coded as in F). Analyses in panels B and F used global signal regression on cortical GCaMP signal to emphasize local networks, this step was removed for all other analysis. (**F**) Correlation maps (zero lag time) between behavioral activity (gradient pixel analysis of videos, see methods) and brain GCaMP-6 calcium-dependent activity made over the 151 concatenated head-fixed sessions. Pixel by pixel brain GCaMP correlation with three different behavioral ROIs are shown. (**G**) Plots of behavioral ROI values on the same timescale as mesoscale GCaMP signal from frontal motor cortex, the gradient values were auto-scaled based on min/max movements. During each cycle of movement that typically involved multiple body parts there was a large GCaMP signal within frontal motor cortex (2nd head-fixed session from 09252018 for M8423, 38–76 s of the dataset), ROI is the anterior red circle in panel B. (**H**) Single trial pseudo colored changes in GCaMP6 fluorescence with during licking task (cue time indicated) for 50 consecutive rewarded Go hit trials that received a code of 2 for mouse 8423 using data from 08252018 (each image reflects the maximal value of 400 ms time bins).

Mesoscale wide-field imaging can inform the areal extent of cortical activation, but the procedure is not without its limitations. Imaging green epifluorescence is sensitive to interference from hemoglobin (both the amount and oxygenation content are factors) (*Ma et al., 2017*; *Ma et al., 2016*; *Wekselblatt et al., 2016*; *Xiao et al., 2017*). While blood dependent alteration in signals is less of a factor for newer generation GCaMPs with larger change in fluorescence divided by baseline (dF/Fo) (*Vanni and Murphy, 2014*; *Xiao et al., 2017*), it can nonetheless be a significant confound. Recent work suggests several correction strategies for green epifluorescence involving reflected light signals that serve as reference wavelengths and are the basis of most corrective schemes (*Ma et al., 2017*; *Ma et al., 2016*; *Wekselblatt et al., 2016*; *Xiao et al., 2017*). Within this improved home cage system we simultaneously capture a short blue wavelength reflected light (440 nm; near the hemoglobin isosbestic point) signal (into the blue RGB channel of the picam) to estimate these and other artifacts and correct for movements that would have parallel effects on short blue reflected light and green epi- fluorescence. The collection of this reference signal was made possible by using a triple band emission filter (*Table 1*). These artifacts were evaluated by determining both green epi-fluorescence and short blue reflected light dF/Fo and subtracting equaled scaled short blue reflection changes from the green epi-fluorescence dF/Fo at each pixel and time point studied. We acknowledge that future extensions could tune the weighting of these signals or add additional wavelengths (*Ma et al., 2017*). Typically, these changes in short blue reflection were <20% of the green GCaMP signal amplitude (*Figure 7G*) and produced relatively small decreases in reflected light signal apparent during averaged task dependent activity (*Figure 8* and *Figure 8—figure supplements 1–7*). Technical considerations associated with using the Picam and its rolling shutter prevented us from collecting a green reflected light signal in alternating frames. While we acknowledge that a multispectral and model based strategy (*Ma et al., 2017*) would be an improvement, the procedure we employ nevertheless correlates well with the green reflected light (*Xiao et al., 2017*) strobing correction as in *Wekselblatt et al., 2016* and was simpler to employ in the context of the home cage. To link mesoscale GCaMP activity with particular phases of a behavioral task, we used the lick detector as a means of registering a behavioral output. Signals on the lick detector were thresholded and recorded as binary events and assigned unix timestamps within a text file which accompanied each imaging session. Using Matlab software, we associated these text file time stamps with sequences of brain images also identified by a starting timestamp and confirmed by measuring the timing of the illumination LED on and off signals. To assess spatial extent of single trial signals we made montages of blue reflectance corrected GCaMP $dF/F_0$ images that were binned over 400 ms (*Figure 7H*). These sequences of $dF/F_0$ images even in single trials indicated remarkable fidelity around the delivery of the stimulus during go-trials and a wide cortical network activated during correctly scored go-trials.

## Go and No-go task headfixed and non-headfixed task acquisition

In a mouse that was trained in the Go and No-go task for 49 and 36 days respectively we created average maps of cortical activity aligned to the presentation of the cue stimulus (*Figure 8A*); for other individual mouse examples see *Figure 8—figure supplements 2* and *3* (mouse 8474 and 0252). In *Figure 8* the mouse was trained to suppress licking for a 2 s +- 0.5 s random interval and then was presented with a vibration to its left paw after which it needed to wait for a delay period (self-timed, 900 ms) and then lick to be given a reward 1.25 s later (as described in *Figure 2B*). *Figure 8* shows an example of average cortical maps of task and reward-dependent mesoscale GCaMP functional signal from cortex during rewarded trials (*Figure 8A,B* and *Figure 8—Video 1*). We observed quiescence within brain networks during the lick withhold period (pre-cue stimulus) when the animals must suppress licking and then a much larger response in sensori-motor cortex associated with preparation (during the delay period after the cue), execution of licking (*Chen et al., 2017*; *Guo et al., 2014*), and collection of the reward. By making averages of the GCaMP data and plotting regions of interest from motor cortex, we also observed changes in GCaMP6 signal before the presentation of the cue. It is possible that some of this brain activity is related to ongoing body movement dynamics associated with the task. To address this, we examined regions of interest from synchronized behavioral cameras directed on the body of the animal. Using the hindlimb behavioral ROI we compared all trials collected to those where hindlimb activity was low for 2 s preceding the stimuli (*Figure 8B*). This constraint led to the elimination of about 40% of the trials and removed a portion of the pre-trial brain activity from the average (see −2.5 and −2 s images, Fig, 8A/B and

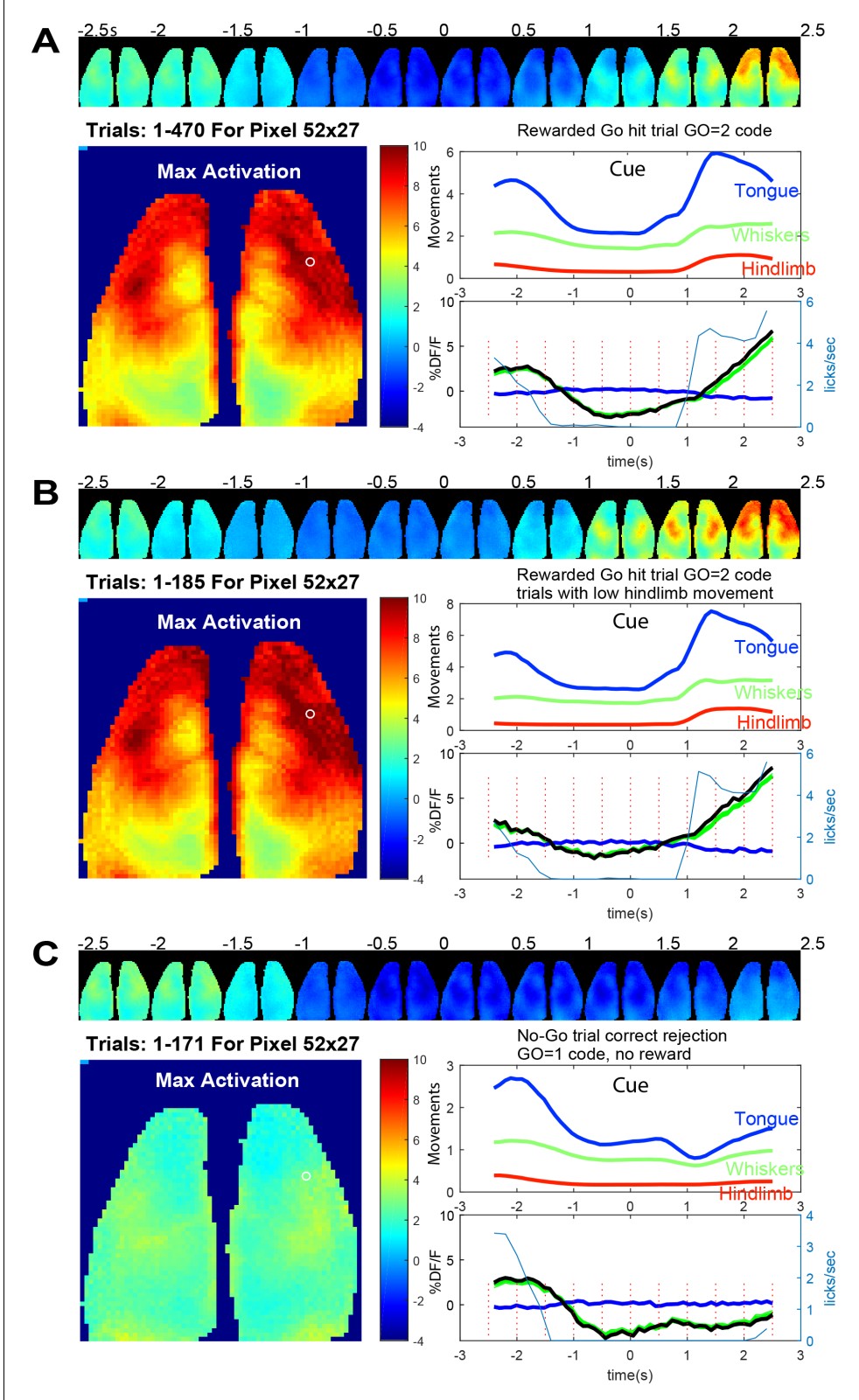

**Figure 8.** Averaged brain GCaMP6 and behavioral ROI data for licking task. (**A**) Trial averaged pseudo-colored GCaMP cortical image montage binned every 500 ms (pixelwise to show maximum values) during go-task activity. Data from a 2.5 s period before and after vibratory go-cue presentation (500 ms vibration, begins at time 0) for mouse tag 8423 and 49 days since the start of the go task training and 36 days from no-go. Large image shows the maximal activation over this period (difference between pre-cue minimum and post-cue maximum values for images binned over 100 ms). Upper panel

*Figure 8 continued on next page*

*Figure 8 continued*

plot normalized behavioral gradient signal for indicated body region ROIs. Bottom panel plots: average GCaMP signal for 470 trials of successful licking Go-trials (lick rate plotted in blue thin lines) for a pixel in anterolateral motor cortex (white circle coordinates 52, 27 in the 64 × 64 image shown, note image was binned spatially from 256 × 256 raw data). By examining the lick rate plot, the lick withhold time (with random 500 ms interval) can be seen when licks plummet to 0 just before trial start. The response window after the self-timed delay (900 ms) is also visible with licks being detected again and then a second peak of licks at the reward delivery is observed. (B) Identical plots as A for trials where hindlimb movements 0–2.5 s before the vibration were <0.4 stdev (ROI 1 = tongue/mouth, 2 = whiskers, and 3 = hindlegs). This criterion makes pre-stimulus baseline activity more constant but otherwise shows similar dynamics. (C) Identical plots as B but for trials where a no-go stimulus (3, 100 ms vibrations with 200 ms inter stimulus interval) was given. These data were autonomously obtained from 9/25/2018 to 9/29/2018 and are the average of all head-fixed trials of given outcome type. The online version of this article includes the following video and figure supplement(s) for figure 8:

**Figure supplement 1.** Averaged GCaMP6 and behavioral ROI data for licking task early training data.
**Figure supplement 2.** Averaged GCaMP6 and behavioral ROI data for licking task early training data from mouse 8474, Go and No-Go.
**Figure supplement 3.** Averaged GCaMP6 and behavioral ROI data for licking task early training data from mouse 0252, Go and No-Go.
**Figure supplement 4.** Averaged GCaMP6 and behavioral ROI data for licking task error trials for dataset presented in *Figure 8*.
**Figure supplement 5.** Averaged GCaMP6 and behavioral ROI data for licking task: error trials for dataset presented in *Figure 8—figure supplement 1*.
**Figure supplement 6.** Averaged GCaMP6 and behavioral ROI data for licking task error trials for mouse 8474.
**Figure supplement 7.** Averaged GCaMP6 and behavioral ROI data for licking task error trials for mouse 0252.
**Figure 8—video 1.** Correct response hit-trial Go-trial (Go = 2 code), averaged pseudo-colored GCaMP cortical images over a 2.5 s period before and after vibratory go-cue presentation for mouse M8423.
https://elifesciences.org/articles/55964#fig8video1
**Figure 8—video 2.** Correct rejection no-response No-Go-trial (Go = 1 code), averaged pseudo-colored GCaMP cortical images over a 2.5 s period before and after vibratory cue presentation for mouse M8423, not rewarded.
https://elifesciences.org/articles/55964#fig8video2
**Figure 8—video 3.** In-correct response Go-trial licked early (−4 code, considered hit because they responded to correct cue), averaged pseudo-colored GCaMP cortical images over a 2.5 s period before and after vibratory go-cue presentation for mouse M8423, not rewarded.
https://elifesciences.org/articles/55964#fig8video3
**Figure 8—video 4.** In-correct response false alarm No-Go-trial (licked in correct time window for No-Go −1 code), averaged pseudo-colored GCaMP cortical images over a 2.5 s period before and after vibratory go-cue presentation for mouse M8423, not rewarded.
https://elifesciences.org/articles/55964#fig8video4

*Figure 8—figure supplement 1*). This finding is consistent with ongoing movements contributing to significant brain activity, a variable which must be addressed in any home cage or head-fixed behavioral task (*Musall et al., 2018*; *Salkoff et al., 2020*; *Clancy et al., 2019*; *Stringer et al., 2019*). This issue was further addressed in a group data analysis present in *Figure 9* where movements before and after the cue strongly correlate with GCaMP dynamics.

The ability to group trials based on task type and outcome allowed us to assess the difference in brain activation between average go and no-go trials in mouse M423 which exhibited good performance during the task (*Figure 8C* and *Figure 8—Video 2*). No-go trials were associated with minimal changes in brain activity and body movements, with most activity around the time of cue processing (1 s after cue presentation) and little activation associated with the advance to licking phase (as they did not lick in correct no-go trials).

In *Figure 8—figure supplement 1* we plot data using the same format as *Figure 8* for an earlier timepoint during training (30 days earlier) for the licking task (after 17 and 4 days of training in the go and no-go task respectively) from the same mouse M8423. In this example similar maps and dynamics were observed, but the self-timed delay period over which post-stimulus licking suppression occurred was shorter. In *Figure 8—figure supplements 2* and *3* we show examples of averaged (correct) Go and No trial data from 2 additional mice from cage 4 (tags M8474 and M0252) and group average data on 13 mice in *Figures 9* and *10* for the Go-detection phase of the task.

## Brain-behavior correlation

Z-scored calcium activity from HL and ALM cortical areas were averaged across trials for each of 13 mice during successful GO trials (*Figure 9A*; we pooled data from all animals that had at least 20 trials). These traces were then used to compute an across-mouse averaged record (*Figure 9B*, black traces). To determine which mice exhibited different cortical activity patterns during the task, the Pearson correlation coefficient was calculated between each mouse's single trial calcium activity and

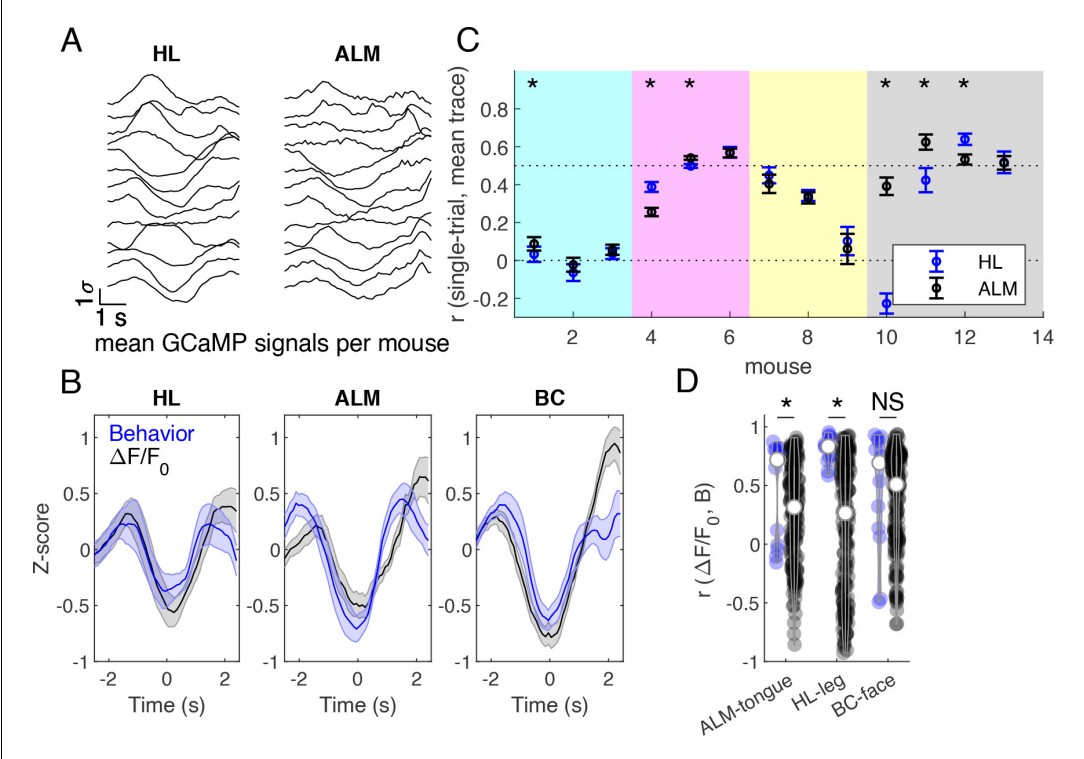

**Figure 9.** Brain-behavior correlations on grouped mouse data during successful Go trials. (A) Z-scored cortical fluorescence values plotted for each of 13 mice, first averaged spatially over the cortical region, then averaged across trials, for hind limb (HL) and anterior lateral motor (ALM) cortical areas. (B) Averaged z-scored fluorescence signal (black) and behavioral gradients (blue) across mice in hindlimb (HL), anterior lateral motor (ALM), and barrel cortex (BC) areas. Mean +/- SEM. (C) Correlation between single trial fluorescence activity and the averaged fluorescence signal across mice in hind limb (blue) and anterior lateral motor cortex (black). Markers indicate mean +/- SEM. Mice from different cages are grouped by background color (females last 4 mice, #10-13). (D) Correlations between average fluorescence signal and behavior gradients. Blue violins show correct pairings, and black violins show mismatched pairings across mice. * indicates p<0.05, Wilcoxon signed rank test.

the across-mouse averaged trace (*Figure 9C*). Some mice exhibited similar activity patterns during the trial (e.g. correlation >0.5, *Figure 9C*), while other mice had a different strategy consistent with recent work (*Gilad et al., 2018*). Surprisingly, the cage of female mice trained using slightly different training parameters in the laboratory had good correlation to the overall GCaMP signal dynamics in the pooled group (*Figure 9C*, see far right points). To compare calcium signal dynamics (*Figure 9B*, black traces) to behavioral activity, the behavior gradient across the tongue, leg, and face areas were z-scored and then averaged across trials (*Figure 9B*, blue traces). The Pearson correlation coefficient was calculated between the behavioral gradient and the GCaMP calcium activity in the corresponding region for each mouse (*Figure 9D*, blue violins). The correlations indicated that a significant amount of covariation in the two signals. Signals from HL and ALM cortical regions were chosen for their association to body movements and licking, respectively. Brain-behavior pairs were then shuffled (across mice), where all pairs of cortical and behavioral activity were considered except for those which include the correct pairing of mice (e.g. correct pair correlation mouseA$_{brain}$ ALM, mouseA$_{behavior}$ tongue). The Pearson correlation coefficient was then calculated for shuffled brain-behavior pairs (*Figure 9d*, black violins). Shuffling brain-behavior pairs across mice resulted in a significant decrease in brain-behavior correlation in HL and ALM regions, but not BC (p<0.05, Wilcoxon signed rank test) suggesting tight coupling between the brain task GCaMP and the movement of tongue or hindlimb body parts.

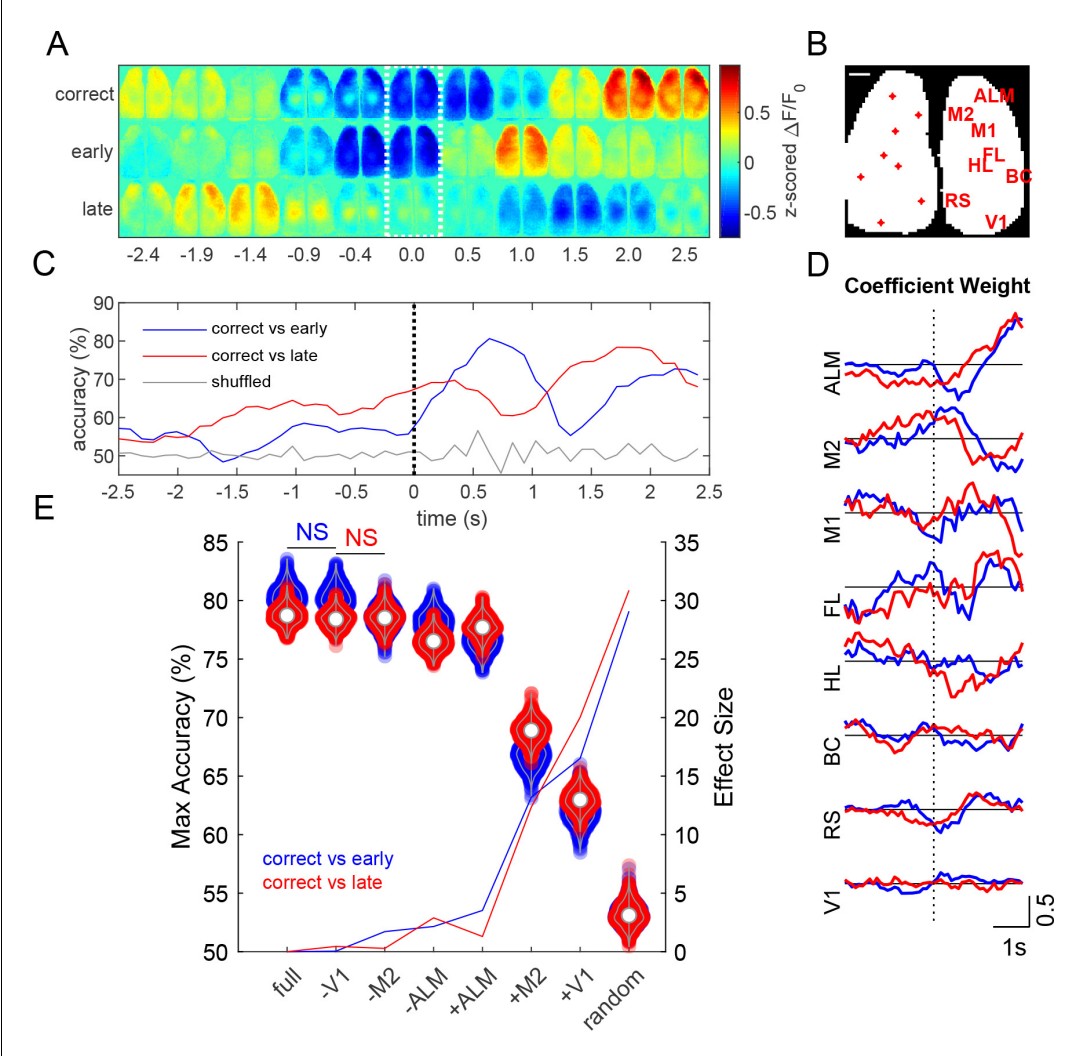

**Figure 10.** Prediction of task outcome from cortical fluorescence signals using a ridge regression model. (**A**) Z-scored fluorescence maps over the course of the trial (averaged across 454 trials averaged over 13 mice for each trial outcome). Dotted region indicates time when cue is delivered. (**B**) Locations of cortical regions used for the general linear model (scale bar: 1 mm). (**C**) Task outcome prediction accuracy vs time. Black dotted line indicates the time when the stimulus was delivered. (**D**) Coefficient weights as a function of time for correct vs early (blue) and correct vs late (red) models. (**E**) Violin plots show maximum accuracy for full and reduced models. Solid lines show effect sizes for reduced models compared to the full model. Model types denoted by (-) indicate that the region specified was randomly permuted, whereas (+) indicates that all regions except for the specified region were randomly permuted (e.g. –V1 indicates V1 calcium signal trials were shuffled, whereas +V1 indicates that all calcium signal trials except those from V1 were shuffled). ($p<0.05$, repeated measures one way ANOVA with post-hoc Bonferroni correction for all groups except those labeled not significant (NS)).

## Decoding of trial outcome

Z-scored $\Delta F/F_0$ montages, averaged across mice (454 single trials over 13 mice, for each trial outcome), show different cortical calcium dynamics associated with each Go-trial outcome: correct (code +2), lick late (code −2), or lick early (code −4) (*Figure 10A*). Activity from various cortical regions were used to predict task outcome using a generalized linear model with ridge regression (*Pinto et al., 2019*). Seed pixel locations and labels are shown over a binary mask outlining the bilateral cortical image (*Figure 10B*). The model achieved a maximum prediction accuracy of approximately 80% (*Figure 10C*). Anterior regions (ALM and M2) generally had greater coefficient weights during the stimulus response period than posterior regions (*Figure 10D*). To examine the effect of specific cortical regions on prediction accuracy, we created reduced models by randomly permuting fluorescence activity of the corresponding region(s) with respect to trial outcome (*Salkoff et al.,*

*2020*). Reduced models with individual regions randomly permuted showed small changes in maximum prediction accuracy (*Figure 10E*, violins), where permutation of ALM trials resulted in the greatest decrease in model accuracy compared to any other individual region (p<0.001, repeated measures ANOVA with post-hoc Bonferroni correction). Similarly, running the model using only ALM fluorescence signals (+ALM) yielded significantly higher accuracy than when using only M2 (+M2) or V1 (+V1) signals (*Figure 10E*, violins, p<0.001, repeated measures ANOVA with post-hoc Bonferroni correction).

## Supplemental task error analysis examples

Using the text file tags, we examined various types of licking related error trials, including trials where a mouse failed to respond to a stimulus (*Figure 8—figure supplements 4A* and *5A* for two timepoints in training for mouse M8423 and *Figure 8—figure supplements 6A* and *7A* for M8474 and M0252). These included no-go errors where the mouse was given a no-go cue, responded by licking, but was not rewarded (GO=-1) (*Figure 8—figure supplements 4B* and *5B* for two timepoints in training for mouse M8423 and (*Figure 8—figure supplements 6B* and *7B* for M8474 and M0252, *Figure 8—Video 4*). Also of interest were trials in which the mouse licked early, during the delay period during a go-trial, and was not rewarded. Analysis of averaged brain images during early response trials (−4 code) indicated that the errors were associated with brain activity during the period prior to the trial, and that this activity was also spilling over into the delay period ((*Figure 8—figure supplements 4C* and *5C* and (*Figure 8—Video 3*). As expected, in lick too early trials, was a robust early increase in frontal cortical activity associated with tongue motor circuits. While mice were found to perform well in the go detection task, discriminating between go- and no-go stimuli was harder for them in practice and was limited to small number of examples and was not the focus of our study. For example, we found that some mice would lick for both go- and no-go trials in response to cues. These mice apparently did not consider a no-go trial as negative reinforcement (time out was small and no overt punishment applied) and merely licked during these trials in addition to the go-trials. In cases where a mouse licked in a no-go trial there was an initial brain response in limb and motor areas but the mouse would fail to show high rates of licking and associated frontal motor cortex brain activity later in the trial as no reward was given (*Figure 8—figure supplements 4B*, *5B*, *6B* and *7B*).

## Discussion

A way to conceptualize brain circuit organization is into micro, meso, and macro length scales (*Bohland et al., 2008*). To enable study of mesoscale circuit interactions, wide field imaging (*Ferezou et al., 2007*; *Kenet et al., 2003*; *Mitra et al., 2018*; *Mohajerani et al., 2013*; *Mohajerani et al., 2010*) permits the definition of functional map-level networks in genetically specified cell populations (*Carandini et al., 2015*; *Gilad et al., 2018*; *Ma et al., 2017*; *Ma et al., 2016*; *Madisen et al., 2015*; *Vanni and Murphy, 2014*; *Zhuang et al., 2017*). Complementing wide field imaging are methods for spatial and temporal control of cortical circuits through the activation of selective opsins (*Ayling et al., 2009*; *Bollu et al., 2019*; *Galiñanes et al., 2018*; *Guo et al., 2015*; *Guo et al., 2014*). An emerging theme is that task-evoked cortical activity is not necessarily locally confined, but is widely disseminating across mesoscale cortical networks (*Allen et al., 2017*; *Chen et al., 2013*; *Ferezou et al., 2007*; *Guo et al., 2017*; *Guo et al., 2014*; *Makino et al., 2017*; *Shimaoka et al., 2019*; *Steinmetz et al., 2019*). The relatively small brain size of the mouse and its relatively transparent skull (*Drew et al., 2010*; *Silasi et al., 2016*) greatly facilitates mesoscale connectivity studies. However, the need for large scale optical access restricts these experiments to headfixed animals as only larger animals such as rats can carry current widefield sensors (*Scott et al., 2018*). While studying cortical mesoscale circuits during active tasks is a subject of many laboratories, training headfixed mice can extend over months (*Gilad et al., 2018*; *Guo et al., 2014*) making home cage approaches increasingly desirable (*Bollu et al., 2019*; *Murphy et al., 2016*; *Silasi et al., 2018*).

We present a versatile system for collection of automated mesoscale brain activity data from voluntarily head-fixed mice. The system allows integration of various data streams including behavioral video and behavioral output using a capacitance-based lick detector. While fully automated rodent in vivo brain imaging is still early in development (*Murphy et al., 2016*; *Scott et al., 2013*), we

anticipate that these systems will have key advantages for the future. These include less animal stress and lower variance due to more consistent animal treatment and experimental design. Recent experimental observations imply that rodent behavioral results can vary based on the sex of the investigator (*Sorge et al., 2014*) confirming that the experimenter perturbs the measurement process. We expect that automated head fixation will remove or at the very least, reduce handler sex bias and other potential procedural biases.

## Group versus single housing and training

In all our experiments we have employed group housing of animals in automated home cages. Group housing has certain advantages in that it is proven to reduce animal stress (*Bartolomucci et al., 2003*), it is cost-effective and creates the potential for higher throughput. Typically, mice in our system were only engaged in head fixation for less than 20 min a day each. Even with 8–10 mice in the cage there still will be enough time for all mice to obtain water without there being competition for this resource. In cases where we have removed poor performing mice, competition at the water spout becomes even less of a problem. One potential improvement for the system would be to physically isolate the mouse undergoing head fixation from the others in the cage (currently the hindquarters is within the open chamber but tail is exposed). We currently have a prototype door system for the head fixing tube which fully encloses the mouse and its tail once automated head fixation is initiated. The door is designed to not injure any animal which is potentially caught at its entrance. While we have designed this partition system, we do not feel it is required at this point as we have lengthened the chamber in relation to the original design helping to isolate the headfixed mouse from its cagemates.

## Optimizing behavioral training for trial number and success rate

In mouse home cages combined with mesoscale imaging we have attempted two kinds of behavioral tasks. The first is the simple go-trial or detection task where mice after a waiting period will respond to a vibratory cue by licking. We find that home cage mice are quite good at learning this task and that this can potentially report activity in circuits associated with the cue detection, as well as motor output associated with licking. We've also employed more advanced training strategies, including a go- and no-go task. In this case we had more variable rates of performance, in particular home cage mice have trouble with suppressing output during no-go trials. We attribute some of the lack of success in no-go training to be due to the nature of home cage training itself. In our home cage task there is no penalty for an animal doing an incorrect trial other than a short time penalty of a few seconds. Because of the potentially aversive nature of the headfixing experience, we decided to not actively punish animals for wrong responses using air-puffs or extended time-outs. A better task for home cage mice to perform would be a discrimination task with two rewarded outcomes, and an animal would specify them by licking from two different water spouts (*Guo et al., 2014*). Alternatively, a lever or wheel could be used to support the choice of 2 rewarded outcomes (*Burgess et al., 2017*). Other ways to motivate mice to do a more difficult task would be to make rewards in some ways more palatable, perhaps adding a choice of regular water versus sugar water. Another scheme might be to limit the total number of trials an animal could perform. With limited trials mice might be more motivated to exhibit a higher success rate rather than the lower rate they exhibit with unlimited time.

The time headfixed for these 23 good male performers averaged 18+−13 min/day (*Table 1*, behavioral data from all mice is available in an online database, see methods). A difference over previous work (*Murphy et al., 2016*) was removing mice from the cage that were poor performers, or insisted on entering the headfixing tube sideways. Other gains in performance might be obtained through mouse specific training criteria. In the current version of the cage software (used for cages 1–5) all animals are advanced through the go and no-go task together. While we report individual mouse results, it was not possible to selectively tailor the task to individual mice. For cage 6 (and the software we will supply going forward) there was the option to advance mice at different rates using animal specific task information. In other work we have done using a home cage lever task in non-headfixed mice, we did have the ability to alter task criteria based on individual animal performance (*Silasi et al., 2018*; *Woodard et al., 2017*). Here, if mice had difficulty with a phase of training, it was possible to downgrade them and make the task easier, allowing them get back on track rather

than left thirsty and requiring supplemental water. The auto-headfixing cage will benefit from such individualized training criteria and it will be instituted in all future cages. While we discuss various improvements in trial design to lead to more complex tasks and even better animal engagement, the current hardware and software is capable of monitoring mesoscale circuits in a behavioral or movement based context. Emphasis in our lab has been around mesoscale imaging, notably the cage is also an ideal platform for mesoscale circuit control using optogenetics as transcranial windows readily support the optical control of targeted brain regions (*Guo et al., 2014*; *Silasi et al., 2013*; *Silasi et al., 2016*).

While we have had success automating headfixing and training, it is clear that some mice are not interested in the system as headfixation is voluntary. In our system we have intentionally kept head-fixed time short, no more than 45 s so that mice will have an opportunity to leave the device. However, during these short trials we find that most mice will re-headfix at a short interval <30 s, indicating a pattern associated with drinking, or some degree of acceptance of the task. It is anticipated that there could optimization of headfixed duration as other work in mice (*Aoki et al., 2017*) did use much longer but fewer blocks of restrained training using a latching mechanism. Other optimization could involve smaller reward sizes (*Aoki et al., 2017*) as we used a ~ 7–10 µl reward and always gave one task-independent entry reward. In this case we would not expect mice to perform more than 120 successful trials to obtain a daily 1 ml total that is typical for mouse training (*Guo et al., 2014*). In our case short bouts of headfixation led to mice performing about 100–200 licking task trials a day (males 24/7 training averaged 200 trials/day, females ~ 7 h/day 100 trials). While 100–200 trials/day/mouse is a much lower than some non-head-fixed home cage tasks where 500–2000 trials/mouse/day were observed (*Bollu et al., 2019*; *Silasi et al., 2018*), one needs to take into consideration that multiple animals are in the same cage and totals can approach 400–800 trials/day when all mice are considered (see *Table 1*). Furthermore, in our case by combining the task with mesoscale imaging each trial needs to have at least 3 s of data collected before and after the cue given the relatively slow GCaMP kinetics making these trials longer than some motor tasks that were less than 2 s (*Silasi et al., 2018*) or evaluated reaching trajectories (*Bollu et al., 2019*).

In summary, we present an open-source tool that is capable of training group-house mice for simple detection tasks while performed mesoscale imaging. We provide an extensive supplemental cage assembly guide and all documentation for hardware, software, and electronics and encourage users to further customize the cage to their needs.

## Materials and methods

### Supplemental material and data availability

We include an extensive supplemental guide with full construction illustrations and parts lists (see Home cageGuide_Murphy_lab_June_2019.docx). As a companion to this guide are CAD files for machined and 3-D printed parts, as well electronics schematics and software for data acquisition and data analysis also present on the journal website. All text file behavioral data are included online as well as image data for *Figure 8* and *Figure 8—figure supplement 1* are found on https://doi.org/10.5281/zenodo.3243572, all data files and code for *Figures 9* and *10* are found in https://doi.org/10.5683/SP2/ZTOPUM and female mouse behavioral data https://doi.org/10.5683/SP2/9RFXRP. All Python data acquisition code can be found on https://github.com/jamieboyd/AutoHeadFix/ (*Murphy, 2020*; copy archived at https://github.com/elifesciences-publications/AutoHeadFix) and https://github.com/ubcbraincircuits/AutoHeadFix.

### Animals

Forty four male and 8 female transgenic C57BL/6 mice expressing GCaMP6s were used. Cage 1 and 3 were Ai94 from Allen Institute for Brain Science, crossed to Emx1–cre and CaMK2-tTA line, Jackson Labs) (*Madisen et al., 2015*), cage 2 Thy-1 GCaMP6s line 4.3 (HHMI Janelia Research Campus) (*Dana et al., 2014*; *Sofroniew et al., 2016*), and cages 4,5, and 6 were tetO-GCaMP6s x CAMK tTA (*Wekselblatt et al., 2016*). These animals underwent surgery, water deprivation/restriction and training as described below. All procedures were conducted with approval from the University of British Columbia Animal Care Committee and in accordance with guidelines set forth by the Canadian Council for Animal Care. Mice were housed in a conventional facility in plastic cages with micro-

isolator tops and kefpt a normal 12 hr light cycle with lights on at 7 AM. When mice were placed in auto-head-fixing cages the bedding was removed and substituted by paper towels to prevent the fine pieces of bedding interfering with the head-fixing or weighing apparatus. The lack of bedding was a precaution; we anticipate that it will be possible to use bedding in the future based on bedding being employed in other similar home cages (*Silasi et al., 2018*; *Woodard et al., 2017*).

## Animal surgery, chronic transcranial window preparation and RFID implantation

Animals were anesthetized with isoflurane and a transcranial window was installed as previously described (*Silasi et al., 2016*; *Vanni and Murphy, 2014*) and in an amended and more extensive protocol described below. A sterile field was created by placing a surgical drape over the previously cleaned surgical table, and surgical instruments were sterilized with a hot bead sterilizer for 20 s (Fine Science Tools; Model 18000–45). Mice were anesthetized with isoflurane (2% induction, 1.5% maintenance in air) and then mounted in a stereotactic frame in with the skull level between lambda and bregma. The eyes were treated with eye lubricant (Lacrilube; www.well.ca) to keep the cornea moist, and body temperature was maintained at 37°C using a feedback-regulated heating pad monitored by a rectal probe. Lidocaine (0.1 ml, 0.2%) was injected under the scalp, and mice also received a 0.5 ml subcutaneous injection of a saline solution containing burprenorphine (2 mg/ml), atropine (3 μg/ml), and glucose (20 mM). The fur on the head of the mouse (from cerebellar plate to near the eyes) was removed using a fine battery powered beard trimmer, and the skin was prepared with a triple scrub of 0.1% Betadine in water followed by 70% ethanol. Respiration rate and response to toe pinch was checked every 10–15 min to maintain surgical anesthetic plane.

Prior to starting the surgery, a No. 1 circular cover-glass (Marienfeld, Lauda-Konigshofen, Germany; Cat#:0111520) was cut with a diamond pen (ThorLabs, Newton, NJ, USA; Cat#: S90W) to the size of the final cranial window (~9 mm diameter). A skin flap extending over both hemispheres approximately 8 mm in diameter (3 mm anterior to bregma to posterior end of skull and down lateral to eye level) was cut and removed. A #10 scalpel (curved) and sterile cotton tips were used to gently wipe off any fascia or connective tissue on the skull surface making sure it was completely clear of debris and dry before proceeding. The clear version of C and B-Metabond (Parkell, Edgewood, NY, USA; Product: C and B Metabond) dental cement was prepared by mixing 1 scoop of C and B Metabond powder (Product: S399), 6 drops of C and B Metabond Quick Base (Product: S398) and one drop of C and B Universal catalyst (Product: S371) in a ceramic or glass dish (do not use plastic). Once the mixture reaches a consistency that makes it stick to the end of a wooden stir stick, a titanium fixation bar (22.2 × 2.7×3.2 mm) was placed so that there was a 4 mm posterior space between the bar edge and bregma, by applying a small amount of dental cement and holding it pressed against the skull until the cement partially dried (1–2 min). With the bar in place, a layer of dental adhesive was applied directly on the intact skull. The precut cover glass was gently placed on top of the mixture before it solidifies (within 1 min) taking care to avoid bubble formation. If necessary, extra dental cement was applied around the edge of the cover slip to ensure that all of the exposed bone was covered, and that the incision site was sealed at the edges. The skin naturally tightens itself around the craniotomy and sutures are not necessary. The mixture remains transparent after it solidifies and one should be able to clearly see large surface veins and arteries at the end of the procedure. In the same surgery a small incision was made over the abdomen and an RFID tag (125 kHz glass Sparkfun) was implanted and secured by a suture in a similar manner as described previously (*Bolaños et al., 2017*). Once the dental cement around the coverslip is completely solidified (up to 20 min), the animal received a second sub-cutaneous injection of saline (0.5 ml) with 20 mM of glucose, and allowed to recover in the home cage with an overhead heat lamp and intermittent monitoring (hourly for the first 4 hr and every 4–8 hr thereafter for activity level). Allow the mouse to rest for approximately 7 days before experimenting with the mouse.

## Camera parameters

To focus the image at ~15 mm from the sample the Picam lens was screwed to its maximal counterclockwise position (threads engaged ~1/2 turn) and was secured with glue. The depth of field was determined in the same way as in previous reports (*Lim et al., 2012*) and was found to be ~3 mm. This provided both a large focal volume over which to collect fluorescence and makes the system

less sensitive to potential changes in z-axis position. To reduce image file size, we binned data at 256 × 256 pixels on the camera for brain imaging data. Brain images were saved in raw 24 bit RGB. We manually fixed camera frame rate to 30 Hz, turned off automatic exposure and auto white balance and set white balance gains to unity. To register frames within and between recording sessions, images were aligned using shifts computed by cross-correlation which was calculated directly (Matlab) or via the FFT to estimate shifts. For both green epifluoresccence and blue reflection channels we adjusted the intensity of illumination so that all values were below 190 out of 256 grey levels (higher levels increase chance of cross talk and saturation). Synchronization between multiple behavioral cameras and brain imaging camera was obtained using a BrainLEDON and BrainLEDOFF text file event which was placed ~3 s from the beginning and end of the headfixed session (duration varies depending on behavioral trial, i.e. the mouse is allowed to finish a trial that has been initiated). Due to small variability in Picam framerates that can accumulate when long sequences of imaging data are concatenated we corrected imported brain and behavioral videos using interpolation in Matlab to yield a framerate of 30 Hz. To assess behavior and movements additional Picams were used that were linked by UDP triggers and also recorded BrainLEDON and BrainLEDOFF text file events to assist in aligning different imaging streams based on interpolation. Regions of interest were drawn over body parts in down-sampled behavioral videos. In the case of HL, this ROI may have also contained contributions to tail or hind quarter movements.

## Training of mice for self-head-fixation

Male or female mice that were at least 45 days of age were used for all experiments (see step by step protocol below and cage building supplemental guide). We performed a surgery to install a chronic transcranial window and head fixation bar (*Silasi et al., 2016*) that was similar to previous windows but without bone thinning (*Drew et al., 2010*). 5–21 days after the surgery, the animals were placed on a schedule of water deprivation and had access to a training cage. Given variation in weight due to ad libitum consumption, the initial mouse weight was defined 24 hr after the start of water restriction. If mice did not progress well through training they were still given up to 1 ml of water daily, there was also a 15% maximal weight loss criterion used for supplementation (see detailed protocol). Once mice have learned the head-fixed task, we allowed them to automatically head-fix themselves and consume water ad libitum. Successfully head-fixing animals were able to maintain body weight and gained weight towards pre-surgery and water restriction values.

## Initial home cage training: optimizing participation, waterspout position, and preventing sideways entry

In initial cages the water spout tip was mounted within 20 mm of the entrance (cages 1 and 2), but we found problems with accurate RFID tag reading since mice would not enter far enough into the imaging chamber to reliably trigger the tag reader. In cages 3–5 the water spout was initially placed 65 mm from the headfixing tube entrance ensuring that animals would come in far enough to have their RFID tag accurately read and that would allow us to confirm that all animals were involved in the initial phases of training and they were able to lick the spout. Because we noticed that some animals would enter the headfixing tube improperly (sideways), we installed removable 3D-printed plastic barriers to prevent an animal with a vertical bar from reaching the water spout if they enter the head fixing tube sideways (see *Table 2*, i.e. AHF_HeadStraightener_50 mm.stl). Two barriers were used as the water spout position was manually advanced in training. The first barrier was 50 mm long and the water spout was placed 15–25 mm after the end of this barrier. Based on the advancing spout positions, we updated mice to a smaller 25 mm barrier and then completely removed the barrier once the water spout is in the final position and it was unlikely that a mouse entering sideways could still reach the spout at this point.

Mice not fully engaged in the task (or continuing to enter the tube sideways) at these early stages of training (first 3 weeks) were removed from the cage. During early phases of training (first 1–3 weeks), we make use of video streaming to continuously monitor the cage and did not require the mice to perform any task to obtain water. By entering the head fixing tube and having RFID tag read, mice received what was termed as an entrance reward. Initially, the volume of rewards would be large (~20 µl) and this amount is gated by the duration of the valve opening (up to 500 ms). Once the mice advance significantly in training, the volume of the entrance rewards can be reduced by

shortening the valve opening time (100–400 ms, drop now ~7–10 µl). We then included additional large task water rewards for breaking the infrared beams which would indicate that the animal has entered the head fixing tube far enough to potentially trigger the headfixation mechanism (these are termed no-fix trials, see below). At this initial stage of testing we also monitored animal weight using the parallel weighing cage technology (*Noorshams et al., 2017*). However, since we individually inspect their health once a day and in the process manually weigh animals, we have relied on manual weights for management of water supplementation for this study. Female mice were trained in automated daily lab-based sessions of ~7 hr duration and lacked auditory feedback for early licking.

## Raspberry Pi and electronics interface

To control the task, we used the Raspberry Pi 3 single board computer. This internet accessible device incorporates HDMI video output and was able to both control the task and capture data. To collect images of mouse entry and exit from the cage, a second Raspberry Pi camera was employed. Assigning each Raspberry Pi a unique IP address allowed us to assess the cages remotely through the internet to monitor whether any animals were in difficulty during chronic imaging or not receiving sufficient water rewards. UDP protocol triggers were used to synchronize behavioral and brain imaging cameras using unique IP addresses (all cameras record a master triggered lights on and off signal as well).

## Electronics

Electronics were produced as described in the schematics present within the supplemental material.

## Ridge regression model

We trained a linear model to predict trial outcome (correct or incorrect GO trials) based on calcium activity recorded during the trial. Incorrect trial outcomes consisted of errors where mice licked during the delay period (before the correct response period) and where they licked after the correct response period (or not at all), hereafter named early and late errors. Calcium activity (454 single trials across 13 mice) from 16 bilateral cortical regions of interest (anterior lateral motor cortex (ALM), secondary motor cortex (M2), primary motor cortex (M1), forelimb area (FL), hindlimb area (HL), barrel cortex (BC), retrosplenial cortex (RS), and primary visual cortex (V1)) was z-scored and then spatially averaged within each region over a $0.7 \times 0.7$ mm area and across hemispheres. To account for multicollinearity, we used a ridge regression model previously described (*Pinto et al., 2019*)[1]. The model was fit with 50 runs of 3-fold cross validation for each frame of calcium activity in the trial. Briefly, the model can be described as:

$$y = \beta_0 + X\beta + \lambda \|\beta\|$$

Where y is a Nx1 vector of outcomes (1 or -1 for correct or incorrect trials respectively) where N is the number of trials, $\beta_0$ is an offset term, X is a NxR matrix of z-scored $\Delta F/F_0$ values where R is the number of ROIs, β is a Rx1 vector of coefficient values, $\|\beta\|$ is the Euclidean norm of the coefficient values, and λ is a penalty term. The optimal value for λ was determined through cross validation (*Pinto et al., 2019*). Prediction accuracy was determined as the proportion of outcomes from the test set that were correctly classified. To evaluate each region's contribution to the model's prediction accuracy, we created reduced models for each predictor (calcium activity from individual cortical region) from the model. Predictors were eliminated by randomly permuting trials of the $\Delta F/F_0$ time series from an individual region (1000 iterations). We also compared the models to a randomly permuted model, in which the trial labels (e.g. correct or incorrect) were randomly permuted (1000 iterations). The difference in prediction accuracy between the full model and the reduced model was the unique contribution of that region's calcium activity to prediction accuracy (*Musall et al., 2019*; *Salkoff et al., 2020*).

## Statistics

Results are presented as mean +- standard deviation unless noted otherwise. Experimenters were not blinded during the experiment or the analysis, although automated acquisition and analysis procedures were employed. Some animals were excluded from the results as indicated (based on poor

head-fixation performance or health concerns) and no method of randomization was used. Sample size measurements for analysis of imaging data were consistent with previous work (*Mohajerani et al., 2013*).

Behavioral events were printed within textfiles with timestamp and RFID tag information and then uploaded to an online MySQL relational database. Raw events were further analyzed in Python and processed into detailed summary information about each chamber entry, head-fix session, task driven trial, lick, earned reward and video file. These newly stored tables were further sourced to generate pipelines for daily monitoring of the animals, statistics and figures, allowing their direct update by adjusting the query sent to the database. The database allows the selection and download of data with SQL queries with libraries like e.g. pymysql (Python) or similar SQL connectors and can be setup locally. For data on cages 1–5 see Zenodo https://doi.org/10.5281/zenodo.3268838 and cage 6 female mice see https://doi.org/10.5683/SP2/9RFXRP, Scholars Portal Dataverse.

Clustering of the behavior was done using 1D kernel density estimation using the timestamp as the variable. Head-fix start times were substituted by a Gaussian with standard deviation of 50 s, which is the average time a mouse needs for a single head-fixation including entering and exiting the chamber. The signal minima that appear between two head-fixes were thresholded at approximately 5% of the peak amplitude of an isolated head-fixation signal to identify the end of a cluster. This corresponds to 2.4 sigma (99% probability) and results in cluster breaks after 4–5 min without head-fixation.

The headfixation task-trial software had provisions to allow mice extra time to finish a partial trial. In a small number of cases (less than 5% of trials) when the last task trial occurred beyond the point of specified headfixation duration there were cases where the trial lacked the usual lick withhold time. In these instances a small number of licks would appear in the lick withhold period. We have removed these trials from the lick analysis and they were typically not included in averaged brain imaging GCaMP data as we only included trials where there was 3 s of data after the stimulus and before the brain LED turns off.

Analysis of brain GCaMP dynamics was performed in Matlab using an analysis pipeline that was similar to that previously reported (*Murphy et al., 2016*), but with the addition of a 440 nm light reflectance image reference subtraction (*Ma et al., 2017*; *Ma et al., 2016*; *Wekselblatt et al., 2016*; *Xiao et al., 2017*). Plots in *Figure 8*, and *Figure 8—figure supplements 2–7* include the mean +-SEM. For analysis of behavioral data we employed ROIs over specific body parts and expressed movement as the absolute value of the gradient between two adjacent frames that are scaled according to the max and min value of all the ROIs.

## Step by step protocol: Auto head fixation: animal care, surgery, training, behavioral task, archiving and analysis. Assemble of a Raspberry Pi-controlled cage is described in the cage assembly addendum (see home cageGuide and *Table 2*)

### Step 1. acclimate group-housed male (or female) mice to large number housing

Note the bulk of the data we report here uses male mice, it is expected that females would train in a similar manner as our previous work used all females (*Murphy et al., 2016*) and training of a small cohort of female mice yielded similar levels of head-fixed training as the males (*Figure 3—figure supplement 1*). Select 8 to 10 male or female mice of C57bl6 background that are post-weaning and pre-pubescent to form a single cage. The goal is to pool different litters to have 8 to 10 animals of similar age (within 2 weeks of each other) and sex for pre-acclimation to a double automatic head fixation training cage. These animals live together within a double training cage for a 2–6 week period prior to transfer to the autoheadfixation cage over which time they undergo and recover from surgery for RFID tags (*Bolaños et al., 2017*) and transcranial windows when they are least 45 days of age (*Silasi et al., 2016*) (see step 2). Potentially aggressive mice will be noted and removed. Note, we have not removed any mice at the double training cage stage (in the 6 cages we report here), but we nonetheless mention this as a potential safeguard.

### Step 2 combined RFID tag and window surgery

Once acclimated to the double cage, the cohort of 8 to 10 mice undergo a combined surgery for both RFID tag implantation and cranial window installation (typically the 8–10 surgeries are performed over 1–3 days). Over this time, the animals are checked every other day for signs of aggression, wounding or barbering, or poor sealing or infection at the margins of their transcranial windows. We employ previously the published protocol from our group (*Silasi et al., 2016*) with the modification of installing a flat titanium bar (McMaster Carr for stock titanium #9039K24) of dimensions as described (*Murphy et al., 2016*) instead of the 4–40 stud bolt. Prior to installation of the bar, the corners are rounded with a file to reduce the chance of a sharp edge catching the walls of the headfixing tube. Cages are configured to use a text message server to message a set of phone numbers when a set duration (for a RFID tag) in the chamber is exceeded in the unlikely, but possible, scenario of a mouse being stranded within the fixation tube. Once the animals have recovered from the surgery (~7–21 days), they are removed as a group from the training cage and placed in the auto-headfixation cage for the start of water restriction.

### Step 3 Non-head fixed water spout training within head fixation cage

Once in the autoheadfixation cage, we remove the water bottle and weigh the mice 24 hr after water restriction. This weight is the baseline for all weight criteria during auto-headfixation training. Any mouse exhibiting weight loss of more than 15% or exhibiting signs of dehydration is supplemented with water and monitored daily for weight, appearance, and behavior (*Guo et al., 2014*). The cage contains a RFID triggered water spout (tip) placed 65 mm from the head fixation tube opening (measured from the opening to the cage) which dispenses water 'Entrance Rewards' upon reading a tag. In this case, it is important to place the water spout and the tag reader so that the RFID tags will be read as the mouse approaches the water spout. In current versions of the cage, we include capacitive lick detection (*Ardesch et al., 2017*) which provides confirmation that the mouse is indeed licking the water spout and has good spout placement for later headfixed trials. An appropriate starting position for the tag reader is centered 20 mm from the headfixing tube entrance; if 'Licks' occur after 'Exits' in the text file, the reader is moved further from the cage (deeper into the tunnel) to prevent periodic loss of tag reading. If the water spout is placed too close to the entrance, the mouse will reach the spout before the tag is read and no water will be dispensed; in this case, with lick detection, licks will be observed on the capacitance sensor but no entries. We recommend investigators use video streaming to confirm appropriate placement of the water spout as the mice enter. During the first day of non-headfixed water spout training, we monitor the reading of RFID tags and confirmation of licking in most mice tested (use Python code stimulator class AHF_Stimulator_Rewards). Any mice not entering the task will be given supplemental water (to a maximum of 1 ml/day, unless they are below the cut-off where the amount can be up to 1 ml plus any deficit) to maintain their weight above the specified limit (85% of baseline). If only one or two mice are not reaching the spout it is best to identify them and mark their tails using black lab marker so they can be easily found and supplemented with water outside of the cage with less disturbance of the other mice. In these other mice water intake can be confirmed automatically using lick detection and automatic weighing (*Noorshams et al., 2017*). Over the next 6–14 days the water spout is moved a total of 40 mm until a reaches a final length of ~7–9 mm from the headfixing chamber back wall or 110 mm from the entrance. At this time first a removable 50 mm and then a 25 mm blocking structure (see CAD parts AHF_HeadStraightener_25 mm.stl and AHF_HeadStraightener_50 mm.stl *Table 2*) were adding to prevent mice from reaching the waterspout when sideways.

### Step 4 beam break training and lick suppression training

Over 7–14 days mice are gradually advanced towards head fixation. For the first 3–8 days mice are trained to move to the end of the chamber with their bar on the overhead rails breaking the IR beam and eliciting the mesoscale brain imaging excitation LEDs to activate (use Python code stimulator class AHF_Stimulator_Rewards). Here mice are given rewards every 5 s for merely breaking the beam and will learn to associate the blue brain imaging LEDs with greater rewards (at this point headfixing probability is at 0). When the mice are going to the tube end the tag reader should be moved so its center is 40 mm from the headfixing tube entrance as the mice advance and the 25 mm blocker (CAD file) used instead of the 50 mm version. When most mice reach this point (typically

within 2–4 days of beam breaking) we begin to train mice to suppress repetitive licking which is the first step in task training. Repetitive licking is flagged by an audible tone: with intervals between licks of less than <1 s met by the tone (use Python code stimulator class AHF_Stimulator_LickWithold-Speaker). Over 2–4 days mice will learn to suppress repetitive licking creating 1–2 s periods without touching the water spout that are then rewarded by water delivery. At this point in training mice are given rewards regardless of subsequent lick timing (they only need to learn to pause licking) and a vibratory cue that marks the end of the no licking period (and later used for go and no-go training) is also introduced. During this time the water sprout is continually manually retracted (and secured with hot melt glue) and its length is measured from the far wall of the head fixation chamber. The final point for water spout placement is 7–8 mm from the back wall and will make it very likely that the mouse will break the infrared beam triggering head fixation (or a no-fix trial) during spout licking. Lick suppression was begun before loose headfixing in cages 3–5, while it was added after in cages 1 and 2.

## Step 5 gradual head fixation task training

The servo motor head fixation system has the ability to partially restrain mice allowing significant movement of their head and movement forward and backwards (15–20 mm) within the head fixation tube. Initially, this loose restraint is given with a random probability of 50%. Within a 7 day period, the travel of the servo head fixation device is gradually increased until the mouse is tightly head fixed at the end of this period (adjust servo travel using the hardware tester found in the main code interface). Adjustments of the servo position should made in small increments as it is possilbe to damage the system by over-travel due to over-heating of the motor or the mechanical force itself. During both partial and full head fixation trials, as well as non-head fixed trials (where the mouse still receives water rewards), we cue blue LED lights which are required for mesoscale imaging at the point of fixation and during the entire trial during in mesoscale GCaMP imaging. These blue lights help the mouse associate the head fixation trial and task with the availability of water. Initially, mice are given rewards for merely triggering a head fixed imaging session. In this case water is dispensed for a 100–500 ms period at regular intervals (every 5–10 s). These initial trials are relatively short in duration being 15–35 s in length. At this initial time mice are not required to do a complex behavior task, but are merely rewarded regularly for being either loose or tightly head fixed. Mice are also rewarded for breaking the infrared beams that trigger the head fixation process during no-fix (non-headfixed) trials. It is important to view the mice on streaming video and to make sure that the water spout is positioned such that they come far enough forward to break the infrared beam. The position of the mouse tongue with respect to the spout needs to be inspected while head fixed to make sure that there are no mice whose head bar placement prevents them from obtaining water when restrained (nose is angled up or down). Such animals need to be removed, or have their bar repositioned. Of note, the infrared beam break can interfere with making behavioral videos using infrared sensitive CMOS cameras. Therefore, we have included electronics that automatically turn off the infrared beam once the mouse undergoes head fixation.

## Step 6 head fixed task training: lick suppression/cued licking

Once mice are exhibiting significant rates of head fixation of (typically >10 min total/day/mouse) we will advance them to a task which can be performed while head fixed. While our initial goal has been to optimize the training for head fixation, we have also addressed whether mice can be trained for a task in the context of the potential stress that the headfixation may bring. A simple detection task was developed to permit the collection of cortical calcium dynamics. A common occurrence is that once head fixed, mice will lick the water spout in expectation of water rewards. As described in Step 4 we train to suppress over-licking and provide auditory feedback to discourage high rates of spout licking and do not present stimuli until a lick suppression period is met. To signal the end of the lick suppression period and that licking will now lead to water delivery we add a vibration cue at this point and reward licking that occurs after at least 200-300 ms after the cue (this is termed the delay period) and before the reward dispensing time. Water rewards are dispensed 1.25–1.5 s after the response withholding interval plus the cue response window (for a 1.25 s water delivery add response period of 0.3 s so reward at 1.55 s) (use Python code stimulator class AHF_Stimulator_LickNoLickDisc).

### Step 7 head fixed training go and No-go task with detection task

Over a period of several weeks animals are advanced into a task once they learn to suppress continuous licking and they are trained to respond to a cue that indicates the end of the lick lockout period (go-trial). We have used a small vibration (that can vary in frequency) of the headfixing tube as the cue for both go and no-go trials. After the cue mice are initially rewarded with water and in a subsequent step are required to lick the water spout after a delay period to receive a reward (as in step 6), for this correct response the software awards a +2 code. We find that head fixed mice have difficulty initially suppressing licking and will lick too early in response to the cue which gives a −4 code or will miss the cue all together −2 code (*Figure 6*). Training mice to delay before responding to the cue is important for future development of a choice task where mice discriminate two related sensory stimuli (in our case we employ two types of vibration a continuous 500 ms or three 200 ms vibrations 100 ms apart). We find that over a period of days we can gradually lengthen this delay period from an initial 300 ms to over 1 s. Correspondingly, we lengthen the period over which mice must suppress licking from 1 s to several seconds (pre-cue period). We also add randomization (+−0.5 s) to the lick suppression interval, so that mice do not learn to rely on internal timing. Over this time, all animals are monitored and the cage is progressed as a group in delay period interval over time. Typically, small increases in delay period are made daily 50–100 ms. The software which runs autoheadfixation reports a series of text codes which indicate whether the mouse licked in response to the cue or not (−2, +2, or −4 described above for Go trials). Once this training is going well a no-go trial is initiated with a different, but related stimulus (pulsed versus continuous vibration). Here the correct response is to withhold licking over the response interval which gives a code of +1, licking during correct time interval (after the stimulus) but during a no-go trial is an error and receives a code of −1, if the mouse licks too early during a no-go trial this a −3 code error (*Figure 6*).

### Step 8 on-going daily data acquisition

Mice are manually weighed as automatic weighing (*Noorshams et al., 2017*), while feasible was not used as the mice were each manually inspected and weighed. Animals are also inspected for any health changes, including ruffled fur, wounding, or signs of dehydration or trauma. Typically, once animals have learned the task, those which headfix require very little supplemental water, whereas non-headfixing animals require daily supplementation of 0.5–1 ml or need to be removed from the cage. We generally find that animals which fail to head fix 2 weeks after the initiation of head fixed training are unlikely to change and adopt headfixation. Secure file transfer protocol (sftp) is used to download text files made daily for each cage, both a detailed record of all events with time stamps and tag numbers, and a brief total for each mouse of head fixes, rewards given, and task performance.

### Step 9 data analysis

Each day automated, cage-based mesoscale imaging can generate an average 15 min of data/ mouse of head fixed mesoscale data acquisition. Taking advantage of the nocturnal nature of mice, we typically back up data during the light cycle. It is important to note that Raspberry Pi external USB mechanical hard disks can drop frames if a hard disk is simultaneously writing new files and being backed up. Using Matlab, we import text files which indicate behavioral performance and licking along with RFID tag and time stamps and use them to parse raw file video of brain activity or H264 files of behavior. Each image file is saved with a timestamp and the RFID tag in the filename. An important step in data analysis is to use a common timing signal that can be viewed in all data streams: text file, behavioral video, and brain video. Typically, we achieve this by making an LED light turn on and off signal. Between lights on and off, we establish the time base for the various camera or text file inputs, this event is also printed within the text file as BrainLEDON and BrainLEDOFF.

## Acknowledgements

This work was supported by a Canadian Institutes of Health Research (CIHR) THM FDN-143209, NIH R21 to THM, M Balbi and T Murphy were supported by a Fondation Leducq team grant, from Brain

Canada for the Canadian Neurophotonics Platform to THM, and a Canadian Partnership for Stroke Recovery Catalyst grant. CIHR or Brain Canada had no involvement in the research or decision to publish. We thank Pumin Wang for help with surgery and Cindy Jiang for animal colony management and Lucas Pinto for providing the code used for the ridge regression model.

## Additional information

### Funding

| Funder | Grant reference number | Author |
|---|---|---|
| Canadian Institutes of Health Research | FDN-143209 | Timothy H Murphy |
| Fondation Leducq | | Timothy H Murphy Matilde Balbi |
| Canadian Neurophotonics Platform | | Timothy H Murphy |
| Michael Smith Foundation for Health Research | | Matilde Balbi |
| National Institutes of Health | R21 | Timothy H Murphy |

The funders had no role in study design, data collection and interpretation, or the decision to submit the work for publication.

### Author contributions

Timothy H Murphy, Conceptualization, Resources, Data curation, Software, Formal analysis, Supervision, Funding acquisition, Validation, Investigation, Methodology, Writing - original draft, Project administration, Writing - review and editing; Nicholas J Michelson, Data curation, Software, Formal analysis, Methodology; Jamie D Boyd, Conceptualization, Data curation, Software, Formal analysis, Methodology; Tony Fong, Software, Formal analysis, Investigation; Luis A Bolanos, Resources, Visualization, Methodology; David Bierbrauer, Resources, Data curation, Software, Formal analysis, Investigation, Methodology; Teri Siu, Software, Formal analysis; Matilde Balbi, Investigation, Drew figures; Federico Bolanos, Conceptualization, Resources, Software, Formal analysis, Investigation, Methodology; Matthieu Vanni, Data curation, Software, Formal analysis, Validation, Investigation, Methodology; Jeff M LeDue, Conceptualization, Resources, Data curation, Software, Formal analysis, Supervision, Methodology, Writing - original draft, Project administration, Writing - review and editing

### Author ORCIDs

Timothy H Murphy ⓘD https://orcid.org/0000-0002-0093-4490

### Ethics

Animal experimentation: All procedures were conducted with approval from the University of British Columbia Animal Care Committee and in accordance with guidelines set forth by the Canadian Council for Animal Care.

### Decision letter and Author response

Decision letter https://doi.org/10.7554/eLife.55964.sa1
Author response https://doi.org/10.7554/eLife.55964.sa2

## Additional files

### Supplementary files

- Supplementary file 1. Illustrated supplemental cage construction guide.
- Supplementary file 2. CAD for 3D printed and machined parts.

- Supplementary file 3. Electronics schematics.
- Transparent reporting form

## Data availability

The name of each brain imaging file which contains both the mouse ID and the time stamp can be found in the SQL database (RFIDtag_xxxx_timestamp.raw; see Methods for URL and hosted as a full text file archive on Zenodo (https://doi.org/10.5281/zenodo.3268838) for the 5 cages of male mice that compose figures 1-7 and cage 6 female mice https://doi.org/10.5683/SP2/9RFXRP. All text file behavioral data is included online as well as image data for Figure 8 and Figure 8—figure supplement 1 are found on https://doi.org/10.5281/zenodo.3243572, all data files and code for Figures 9 and 10 are found in https://doi.org/10.5683/SP2/ZTOPUM and female mouse behavioral data https://doi.org/10.5683/SP2/9RFXRP. All Python data acquisition code can be found on https://github.com/jamieboyd/AutoHeadFix/ (copy archived at https://github.com/elifesciences-publications/AutoHeadFix) and https://github.com/ubcbraincircuits/AutoHeadFix.

The following datasets were generated:

| Author(s) | Year | Dataset title | Dataset URL | Database and Identifier |
|---|---|---|---|---|
| Murphy TH | 2020 | Homecage data for Figures 8 and 9 and behavioral database | https://doi.org/10.5281/zenodo.3243572 | Zenodo, 10.5281/zenodo.3243572 |
| Murphy TH | 2020 | Code and. mat files for automated homecage paper (Figs 9 and 10) | https://doi.org/10.5683/SP2/ZTOPUM | Scholars Portal Dataverse, 10.5683/SP2/ZTOPUM |
| Murphy TH | 2020 | Home cage data female mice cage 6 all text file and database information | https://doi.org/10.5683/SP2/9RFXRP | Scholars Portal Dataverse, 9RFXRP |

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
