## [Decision Letter]

**Acceptance summary:**

Murphy and co-workers have advanced the field of high throughput imaging coupled with behavior to a new level of automation – demonstrating around 2,000 self-initiated imaging sessions per day per mouse for all mice in a colony- that incorporates an online SQL database to segue with the ambitions of a multitude of community-wide efforts in data sharing. This work significantly extends the 2013 pioneering effort of the Brody and Tank collaboration, as well as prior work by the Benucci, Goldberg and Murphy laboratories, among others, to create effective, efficient, and open source tools for longitudinal studies of the neuronal basis of behavior.

**Decision letter after peer review:**

[Editors’ note: the authors submitted for reconsideration following the decision after peer review. What follows is the decision letter after the first round of review.]

Thank you for submitting your work entitled "Automated task training and longitudinal monitoring of mouse mesoscale cortical circuits using home cages" for consideration by *eLife*. Your article has been reviewed by three peer reviewers, and the evaluation has been overseen by a Reviewing Editor and a Senior Editor. The following individual involved in review of your submission has agreed to reveal their identity: James Ackman (Reviewer #3).

Our decision has been reached after consultation between the reviewers. Based on these discussions and the individual reviews below, we regret to inform you that your work will not be considered further for publication in *eLife*.

All three reviewers of your "Tools and Resources" article, as well as the Reviewing Editor, are serious players in the field of cortical dynamics and imaging from awake animals. They and I recognize the continued impact of your work on automated behavior and imaging with modestly priced equipment to promote high throughput experiments and enable adventurous queries. Yet the content in the current paper, which includes technical advances and demonstration imaging data, is judged to be too incremental to warrant publication. The claims will gain in impact if the increase in throughput is used to solve a narrow but nontrivial scientific issue, or make a novel kind of observation, as opposed to making a solely demonstration measurement.

We kept in mind the criteria for the Tools and Resources category in reaching this decision. This category highlights tools or resources that are especially important for their respective fields and have the potential to accelerate discovery. Tools and Resources articles do not have to report major new biological insights or mechanisms, but it must be clear that they will enable such advances to take place. The concensus was that, while there are many laudable aspects to your approach, the current approach is not a sufficiently large advance over the previous approach to warrant publication in *eLife*.

We would welcome a new research article that involves an application of this new tool. The Reviewing Editor provided some suggestions, but of course, we recognize that you are already engaged in a vigorous research program that might encompass these or serve as an alternative. I pass them along for your consideration.

1) An expanded imaging study during the Go/NoGo task, i.e., the use of many mice to establish cortical patterns instead of the one mouse as shown in Figures 8 and 9.

2) Activation of the orofacial regions (forepaw, jaw, tongue, vibrissa) in motor and sensory cortices are certainly of interest.

3) An imaging study from multi-animals study that looks at potential gender differences, with reasonable statistical bounds and controls, could fulfil this role.

4) Other extensions, e.g., plasticity of cortical responses during learning, may be more feasible.

Reviewer #1:

In this "Tools and Resources" article, the authors report on an improved methodology for doing widefield imaging in the home cage. The opportunity to measure behavior and neural activity is important, and the authors highlight reasons such as reduced stress for the animal, and removal of experimenter-to-experimenter variability. Further, the chance for automated fixation in conjunction with widefield imaging is exciting, as widefield imaging is turning out to be an informative and high-throughput way to measure neural activity during behavior.

The method proposed here is an improvement over a previous automated head-fixing apparatus that the authors reported a few years ago. Here, the mice are willing to have daily fixation durations that are much longer. In about half of the mice, the observed 28+/-17 headfixations/day (18+/-13 mins/day).

The paper has a lot of strengths. It includes data from 44 mice and a total of >29K task related videos. The authors also created a relational database to allow analysis pipelining. This information, along with a lot of other useful information, such as the drawings for the head straightener that prevents the mouse from going in sideways, are all provided. There are also a lot of nice touches in the system, such as the text messaging system that alerts the experimenter if animals are in the tube too long; this is thoughtful from the point of view of animal welfare.

There were also a number of observations based on this large-scale approach that were interesting. These include the linked behaviors between mice (Figure 4A, B) and the fact that mice had higher success rates while headfixed although they preferred to be head free (Figure 5C). The fact that performance was similar over the 24-hour cycle was also interesting.

Finally, to deal with hemodynamic contamination of the widefield data, the authors captured 440 nm light in addition to the GCaMP signal. They then subtracted one from the other. The authors acknowledged the really beautiful alternative methods for this developed by the Hillman lab, but justified their use of a simpler method based on some constraints of the device, which I thought was fine. Other papers in the literature have entirely omitted hemodynamic correction so I appreciated the efforts to do this.

First, I was concerned that the authors didn't make sufficiently clear how this work is an advance on the Aoki, Benucci et al. approach that was published in 2017. The text states that, "While an advance for training, this work did not longitudinally gather brain imaging data in an autonomous manner, nor were the systems of a footprint or cost appropriate for running at scale." But in the Aoki paper, 2-photon data was collected for 21 days (Figure 5D). The measurements were not taken in the home cage; if the authors feel that is a critical difference, they should state why. I also wasn't sure how to compare the cost difference with that approach and the one presented here. The Aoki apparatus does appear to be commercially available (http://ohara-time.co.jp/wordpress/wp-content/uploads/SelfHead-restraintOperantTestSystem_Pamphlet2017.pdf), but I wasn't sure of its price or the total price of the setup in the current paper so it was hard to compare.

Second, the authors included no female mice in their study. There seems to be no reason that a cage of all female mice might be tested as well. Inclusion of both sexes in a study is now required by many funding agencies, and is good scientific practice because it ensures that scientific conclusions are not made that only apply to one sex and not the other.

Third, I wasn't sure whether the total fixation time of the mice provided sufficient time for measurement of behavior and neural activity. It seemed that the total fixation time wasn't very long – on order of about 18±5 minutes/day (subsection “Improvements to the cage hardware”) for the good performers. It would be helpful to see the distribution of total daily head-fix times for all mice, as well as the distribution of within-day times, at least for some of the good mice. It would also be helpful to see the distribution of total daily completed trials. The Bollu paper (which admittedly isn't a totally fair comparison) reported >2000 trials/day and the Aoki paper reported ~1000 trials/day. The reason I bring this up is that widefield measurements can be very noisy and so high trial counts are extremely helpful. This is especially true given that, as the authors point out, the signals reflect movements, the timing and magnitude of which vary a lot from trial-to-trial. If the approach in this paper is to be useful for sensory or cognitive tasks, large trial counts are needed.

A related point, I couldn't quite understand the sentence comparing good and bad performers (see the aforementioned subsection). It stated that bad performers had less than 20 min of head-fixed time while good performers had while "good performers averaged 15.8+-19.9 h." I think h can't mean hours since the animals didn't headfix 16 hours/day. But it can't mean minutes either, because that would imply that the good performers were fixating less than the poor performers.

Finally, I found that the behavioral/widefield data included at the end (Figure 8 and the section titled, "Task training error analysis examples") was not very useful. The problem is that it is hard to conclude very much from such a small sample size. There is only 1 animal shown in Figure 8 and so the comparison between go and no-go trials (Figure 8B vs. Figure 8C) is not that informative. Further, the data didn't really add that much to the paper. The reason is that this paper is meant to be a tool and a resource. It does that well. The scientific observations in Figure 8 weren't really in the service of that goal. It might be better to include that (potentially interesting) data in a different paper with a larger cohort of subjects and a lot more analyses.

Reviewer #2:

Achieving high-throughput behavioral training and subsequent automated task administration with physiology is a significant aim for the field of systems neuroscience. The Murphy lab and others have made notable contributions to this aim, which are described in the Introduction to this manuscript. The Murphy group, specifically, has previously reported automated methods for home cage behavioral training and for head-fixation with mesoscopic calcium imaging (in the absence of a specific behavioral paradigm). Here, they describe improvements on their previous head-fixation and imaging method and integration with behavioral training and task performance.

Small improvements in the design of the head-fixation system and refinement of their training protocol yield longer and more frequent head-fixations. Using their updated system, they successfully train 21/44 mice to undergo head-fixation to obtain water rewards. While head-fixed, the mice are trained to perform a detection task with varying degrees of success. Further, 10 mice progressed to a go/no-go task, but only 5 of those mice were able to perform the task (as assessed by a relatively lenient d' criterion), and there is no detailed presentation of this behavioral data in the manuscript. The mesoscopic imaging data presented reflects examples from a few selected mice without any analysis of results across mice, and the major benefits of the system (longitudinal mesoscopic imaging across training) are completely unexplored.

The challenge with this manuscript is that it does not present a compelling methodological result, nor does it describe a compelling experimental finding (it's neither fish nor fowl). The methodology is very promising and the authors should be applauded for the impressive level of detail with which they describe the construction of their system, however the methodology as presented appears to be an incremental improvement on the Murphy lab's previously published results (Murphy et al., 2016), and it's very difficult to tell how much of an improvement it actually represents. The first sentence of the Abstract notes a '4X improved automated methodology', but 4X of what? On the other hand, if the manuscript were focused on an experimental finding or data set, it falls short because of the rather incomplete data (one mouse in many experimental groups, little group level or longitudinal analysis), and further the somewhat opaque manner in which data are presented make it difficult to assess the results and compare to previous work by this group and others.

1) The criterions employed, if any, to move mice between training stages are not described, hampering the reproducibility of the study.

2) It is unclear how the length of individual head-fixations is determined. Can mice exit trials at any time? Further, there are no behavioral measures (other than repeated, "clustered" head-fixation) that indicate whether mice are comfortable in the head-fixation system. A lack of comfort could underlie the poor performance of many of the mice in the detection task.

3) The study pools data from several iterations of the home cage system with different conditions across the cages. Further, cage 3 experienced an unexplained outage. This makes the presentation of data in Figure 3 confusing and difficult to interpret. Clarity would be greatly improved by reporting results for a group of cages with consistent and optimal conditions. At the very least, data should be separated clearly by mouse to make it easier to assess changes in performance. Further, the 23 mice that perform well in training should be presented separately from those that did not progress. Additionally, given that the training periods were of variable length, it is not possible to assess the data (as currently presented) relative to the different training milestones.

4) The inclusion of data from non-head-fixed trials detracts from the manuscript. Given that the system is designed for imaging during behavior (which necessitates head-fixation), only data for head-fixed trials should be presented. There is no comparison of behavioral or imaging data collected with automated (infrequent, short-bout) head-fixation and extended manual head-fixation for comparison.

5) Group-level and longitudinal analysis of behavioral data from the detection task (which should have sufficient N), would strengthen the manuscript. The inclusion of limited go/no-go behavioral data detracts from the manuscript, as it does not appear that the current training regimen has been optimized for performance of this task.

6) In section 1, the authors claim that use of a Raspberry Pi camera module is not inferior, but they do not present any data to support this claim.

7) The mesoscopic imaging presented is sufficient to demonstrate that imaging can be performed in their home cage system. However, that was already established by their previously published manuscript (Murphy et al., 2016). At the very least, group level or longitudinal analysis should be performed to facilitate comparison with results published by other groups with conventional imaging paradigms.

Reviewer #3:

The authors improved upon a home cage system they reported on previously in Murphy et al., 2016. Docking system was improved for greater mouse participation, with a more effective training period. Additional monitoring of animal's behavioral state was added. Functional data is provided for a go/no-go licking task.

All materials are provided, along with schematics and acquisition code, and are a benefit to the greater research community. Automated longitudinal population studies are important for furthering our understanding of the neural basis of behavior.

1) Much of this work has already been published (Murphy et al., 2016). Details provided here are improvements upon the original methods, rather than new findings or techniques.

2) Home cage system is described to be optimized for capturing long-term functional changes in cortex, yet the data showed no functional changes after training. Described results did not elucidate benefits of using this system.

---

## [Author Response]

[Editors’ note: the authors resubmitted a revised version of the paper for consideration. What follows is the authors’ response to the first round of review.]

Reviewer #1:[…] First, I was concerned that the authors didn't make sufficiently clear how this work is an advance on the Aoki, Benucci et al. approach that was published in 2017. The text states that, "While an advance for training, this work did not longitudinally gather brain imaging data in an autonomous manner, nor were the systems of a footprint or cost appropriate for running at scale." But in the Aoki paper, 2-photon data was collected for 21 days (Figure 5D). The measurements were not taken in the home cage; if the authors feel that is a critical difference, they should state why. I also wasn't sure how to compare the cost difference with that approach and the one presented here. The Aoki apparatus does appear to be commercially available (http://ohara-time.co.jp/wordpress/wp-content/uploads/SelfHead-restraintOperantTestSystem_Pamphlet2017.pdf), but I wasn't sure of its price or the total price of the setup in the current paper so it was hard to compare.

We have revised the statement around the Aoki paper and no longer mention cost as an issue as our system is open source and all parts are listed. “Automation has been extended to complex headfixed visual tasks with a relatively good success rate in mice (Aoki et al., 2017) and headfixed rats (Scott et al., 2013). […] Such supervised imaging following home cage training of individual animals will permit high-resolution microscopy in the context of extended automated training (Aoki et al., 2017).” The device sold by Ohara-Time is $32,000 USD/cage based a quote I just received, typically only 2 mice train in one of these cages at a time (not up to 8).

Automation of behavioral and imaging experiments is of keen interest to many in the field and has distinct advantages from an animal welfare standpoint as it may reduce variability leading to the use of fewer animals. We have produced a paper which describes an improved system incorporating many new features (over published work): including an online SQL database, lick detection, auto weight measurement, triggered behavioral cameras, and integration of mesoscale imaging during behavioral task. These features were not present in the original version. Furthermore, published work by others does not support multiple animals within the same cage identified by RFID. There is also little published data on headfixation across mice statistics (now at n=52 mice for our work) in the previous Aoki et al., 2017 work, only that it had been applied, but no daily headfixing records or stats on uptake for specific mice.

Second, the authors included no female mice in their study. There seems to be no reason that a cage of all female mice might be tested as well. Inclusion of both sexes in a study is now required by many funding agencies, and is good scientific practice because it ensures that scientific conclusions are not made that only apply to one sex and not the other.

We now include a cohort of 8 female mice that are presented as Figure 3—figure supplement 1. These mice were successfully trained in the lab during daily 8 h unsupervised sessions (rather than 24/7 animal facility training). Data from these mice are now in group mesoscale GCAMP analysis in figures 9 and 10.

From Materials and methods, “Step 1. Acclimate group-housed male (or female) mice to large number housing; note the bulk of the data we report here uses male mice, it is expected that females would train in a similar manner as our previous work used all females (Murphy et al., 2016) and training of a small cohort of female mice yielded similar levels of head-fixed training as the males (Figure 3—figure supplement 1).” In total we show data from a total of 52 mice that went through this procedure and include the SQL database online with all behavioral data available. Female mice show similar levels of engagement.

Third, I wasn't sure whether the total fixation time of the mice provided sufficient time for measurement of behavior and neural activity. It seemed that the total fixation time wasn't very long – on order of about 18±5 minutes/day (subsection “Improvements to the cage hardware”) for the good performers. It would be helpful to see the distribution of total daily head-fix times for all mice, as well as the distribution of within-day times, at least for some of the good mice. It would also be helpful to see the distribution of total daily completed trials. The Bollu paper (which admittedly isn't a totally fair comparison) reported >2000 trials/day and the Aoki paper reported ~1000 trials/day. The reason I bring this up is that widefield measurements can be very noisy and so high trial counts are extremely helpful. This is especially true given that, as the authors point out, the signals reflect movements, the timing and magnitude of which vary a lot from trial-to-trial. If the approach in this paper is to be useful for sensory or cognitive tasks, large trial counts are needed.A related point, I couldn't quite understand the sentence comparing good and bad performers (see the aforementioned subsection). It stated that bad performers had less than 20 min of head-fixed time while good performers had while "good performers averaged 15.8+-19.9 h." I think h can't mean hours since the animals didn't head fix 16 hours/day. But it can't mean minutes either, because that would imply that the good performers were fixating less than the poor performers.

We now provide the successful trial number for all mice in Table 1 for the last 5

days of training. Typically, 5-6 trials are done in a 40 sec session of headfixation so on average mice were presented with about 100 trials/day. The average trials complete per day is now added in a new column in Table 1. We also have added discussion of trial number, see below.

While we appreciate the elegant design of the Aoki et al., 2017, there is only data shown on n=2 mice in that paper (there is mention that 8 of 12 mice learned the visual task) and no detailed presentation of trial number data for individual mice could be found.

We have also clarified the statement about total time of headfixation. The number in hours was the summed time headfixed over all sessions. “In the current cage the cutoff for a poor performer was <20 min of total headfixation time, while good performers averaged 15.8+-19.9 h (summed time over all daily sessions).”

We have added the following discussion of headfixed trial number “While we have had success automating headfixing and training, it is clear that some mice are not interested in the system as headfixation is voluntary. […] Furthermore, in our case by combining the task with mesoscale imaging each trial needs to have at least 3 s of data collected before and after the cue given the relatively slow GCaMP kinetics making these trials longer than some motor tasks that were less than 2 s (Silasi et al., 2018) or evaluated reaching trajectories (Bollu et al., 2019)”

Finally, I found that the behavioral/widefield data included at the end (Figure 8 and the section titled, "Task training error analysis examples") was not very useful. The problem is that it is hard to conclude very much from such a small sample size. There is only 1 animal shown in Figure 8 and so the comparison between go and no-go trials (Figure 8B vs. Figure 8C) is not that informative. Further, the data didn't really add that much to the paper. The reason is that this paper is meant to be a tool and a resource. It does that well. The scientific observations in Figure 8 weren't really in the service of that goal. It might be better to include that (potentially interesting) data in a different paper with a larger cohort of subjects and a lot more analyses.

There seems to be conflicting responses from the editor and the reviewers since we

understand that the paper was likely rejected as there was not clear group data showing a finding that was supported by mesoscale imaging. The intention of the paper was to: introduce a cage with additional features, detailed instructions on how to build it, describe the training of animals, and show example data to prove that it works and can be used to derive some reported task-dependent activity. We have removed Figure 9 to Figure 8—figure supplement 1 as this was showing a single mouse at 2 training time points and have now add 2 new figures with group data from 13 mice total for the Go detection task.

The new group data figures show that different mice have unique regional patterns of cortical activation during this task and show strong correlations between movements and mesoscale GCAMP signals (Figure 9, subsection “Go and No-go task headfixed and non-headfixed task acquisition”). Also, consistent with recent data is the finding that individual mice have different strategies for task completion. This work also consistent with findings from the Helmchen lab (Gilad et al., 2018). In total this new group analysis of functional imaging data suggests that our home cage is a well-characterized system that can produce results linking mesoscale networks to behavior. The mesoscale GCAMP signals are also used to predict single trial-outcome using a model-based regression approach (Figure 10). Furthermore, the work provides information on the relative contribution of different cortical areas to the predictive value in the regression model extending very recent findings from multiple labs to the context of the home cage (Figure 10). We have also provided group data on the relationship between body movements and mesoscale brain activity. Here, consistent with other findings, there is a strong correlation between movement and mesoscale activity.

Reviewer #2:[…] The challenge with this manuscript is that it does not present a compelling methodological result, nor does it describe a compelling experimental finding (it's neither fish nor fowl). The methodology is very promising and the authors should be applauded for the impressive level of detail with which they describe the construction of their system, however the methodology as presented appears to be an incremental improvement on the Murphy lab's previously published results (Murphy et al., 2016), and it's very difficult to tell how much of an improvement it actually represents. The first sentence of the Abstract notes a '4X improved automated methodology', but 4X of what? On the other hand, if the manuscript were focused on an experimental finding or data set, it falls short because of the rather incomplete data (one mouse in many experimental groups, little group level or longitudinal analysis), and further the somewhat opaque manner in which data are presented make it difficult to assess the results and compare to previous work by this group and others.

We have removed the statement of a 4x improvement in headfixation from the Abstract and now present this argument in the Results only.

“The time headfixed for these 23 good performers averaged 18+-13 min/day (Table 1, behavioral data from all mice is available in an online database, see Materials and methods). […] Assuming a session length of ~40 sec, which was the typical range in the previous system (Murphy et al., 2016), taking the top 50% of headfixing mice would result in an average of 6.5+-4 min of daily data acquisition or training per mouse using the older design.”

1) The criterions employed, if any, to move mice between training stages are not described, hampering the reproducibility of the study.

Admittedly, the system was still under development during the testing of these 44 (plus 8 female) mice so we did not have strict criteria for advancement between stages. In general, we waited for the majority of mice to acquire a particular level i.e. entries into the fixation tube and drinking before advancement. The first version of the software used for cages 1-5 and Murphy et al., 2016 required that all mice be subjected to the same conditions as a group i.e. headfixed or not, or advance to Go and No-Go. The training protocol for cages 4-6 is largely invariant. For the last cohort of female mice we did develop software so that each mouse can have individual training parameters that will allow for future use of automated advancement as we have done in another home cage task with an albeit less complex lever task (Silasi et al., 2018).

2) It is unclear how the length of individual head-fixations is determined. Can mice exit trials at any time? Further, there are no behavioral measures (other than repeated, "clustered" head-fixation) that indicate whether mice are comfortable in the head-fixation system. A lack of comfort could underlie the poor performance of many of the mice in the detection task.

There was no mechanism to automatically release mice using force sensing as all trials were short in duration. Trial length was set by the investigator and generally fixed in the range of no more than 45 sec of headfixation. The trial lengths did vary as we added extra time to allow mice to finish a trial once they started.

From the Discussion section “While we have had success automating headfixing and training, it is clear that some mice are not interested in the system as headfixation is voluntary. […]Furthermore, in our case by combining the task with mesoscale imaging each trial needs to have at least 3 s of data collected before and after the cue given the relatively slow GCAMP kinetics making these trials longer than some motor tasks that were less than 2 s (Silasi et al., 2018) or evaluated reaching trajectories (Bollu et al., 2019).”

3) The study pools data from several iterations of the home cage system with different conditions across the cages. Further, cage 3 experienced an unexplained outage. This makes the presentation of data in Figure 3 confusing and difficult to interpret. Clarity would be greatly improved by reporting results for a group of cages with consistent and optimal conditions. At the very least, data should be separated clearly by mouse to make it easier to assess changes in performance. Further, the 23 mice that perform well in training should be presented separately from those that did not progress. Additionally, given that the training periods were of variable length, it is not possible to assess the data (as currently presented) relative to the different training milestones.

Within Figure 3 we show data from all cages with a legend to indicate cage ID and tag number, we also include a database of all results. We also add a new cohort of female mice and do plot their cage separately in a new supplementary figure, Figure 3—figure supplement 1. We also stress that we include a full SQL database of all events allowing individual mice to be examined. All behavioral data (licking) and functional imaging data is from the 23 mice that performed well and an additional 5 female mice. In the revised group data Figure 9 we now show aggregate as well as individual mouse data and cages data relating body movements to GCAMP signals.

4) The inclusion of data from non-head-fixed trials detracts from the manuscript. Given that the system is designed for imaging during behavior (which necessitates head-fixation), only data for head-fixed trials should be presented. There is no comparison of behavioral or imaging data collected with automated (infrequent, short-bout) head-fixation and extended manual head-fixation for comparison.

We agree that it would be interesting to compare automatically headfixed and extended manual headfixation. However, this was not our intention. Any data regarding the performance in headfixed and non-headfixed cases was included as it might aid someone attempting this work in the future. We should stress that all analysis of mesoscale GCAMP dynamics and task performance is done using fully headfixed trials. We did not use non-headfixed trials in behavioral measures that are presented in figures (such as licking data in Figure 5), other than the comparison of task performance in non-headfixed states.

5) Group-level and longitudinal analysis of behavioral data from the detection task (which should have sufficient N), would strengthen the manuscript. The inclusion of limited go/no-go behavioral data detracts from the manuscript, as it does not appear that the current training regimen has been optimized for performance of this task.

We now add group level analysis in Figures 9 and 10 see these new Figures and the Results section This additional analysis is described below in the response to point 7.

“Brain-behavior correlation

Z-scored calcium activity from HL and ALM cortical areas were averaged across trials for each of 13 mice during successful GO trials (Figure 9A; we pooled data from all animals that had at least 20 trials[…] Scrambling brain-behavior pairs across mice resulted in a significant decrease in brain behavior correlation in HL and ALM regions, but not BC (p<0.05, Wilcoxon signed rank test) suggesting tight coupling between the brain task GCaMP and these body parts.”

“Decoding of trial outcome

Z-scored ΔF/F0 montages, averaged across mice (454 single trials over 13 mice, for each trial outcome), show different cortical calcium dynamics associated with each Go-trial outcome: correct (code +2), lick late (code -2), or lick early (code -4) (Figure 10A). […] Similarly, running the model using only ALM fluorescence signals (+ALM) yielded significantly higher accuracy than when using only M2 (+M2) or V1 (+V1) signals (Figure 10E, violins, p<0.001, repeated measures ANOVA with post-hoc Bonferroni correction).”

6) In section 1, the authors claim that use of a Raspberry Pi camera module is not inferior, but they do not present any data to support this claim.

We have removed the claim that the Raspberry Pi camera is not inferior and have written this in a more conservative manner.

From Results “wide field imaging is typically done with larger focal volume (~2-3 mm) to image curved brain surfaces and lower spatial resolution (pixels binned to ~ 32-49 μm) to sum more photons over a larger area (Lim et al., 2012). Thus, the intrinsically large depth of field and reduced resolution provided by the small lenses employed by compact cameras such as the Raspberry Pi Camera Module (Figure 1B, C) are not expected to degrade signals.”

7) The mesoscopic imaging presented is sufficient to demonstrate that imaging can be performed in their home cage system. However, that was already established by their previously published manuscript (Murphy et al., 2016). At the very least, group level or longitudinal analysis should be performed to facilitate comparison with results published by other groups with conventional imaging paradigms.

In the revision we include group analysis of functional imaging data from a total of 13 mice. For simplicity, we analyze group data from a simple go-detection task and were able to create a model that predicts behavioral trial outcome based on brain activity data (Figure 10). This model was used to examine temporal features and regions which exhibited predictive value in the behavioral task. This is new data and is now presented in two additional figures. We have also provided group data on the relationship between body movements and mesoscale brain activity (Figure 9). Here, consistent with other findings, there is a strong correlation between movement and mesoscale activity. Also, consistent with recent data is the finding that individual mice have different strategies for task completion. This work also consistent with findings from the Helmchen lab (Gilad et al., 2018). In total this new group analysis of functional imaging data suggests that our home cage is a well-characterized system that can produce results linking mesoscale.

Reviewer #3:The authors improved upon a home cage system they reported on previously in Murphy et al., 2016. Docking system was improved for greater mouse participation, with a more effective training period. Additional monitoring of animal's behavioral state was added. Functional data is provided for a go/no-go licking task.All materials are provided, along with schematics and acquisition code, and are a benefit to the greater research community. Automated longitudinal population studies are important for furthering our understanding of the neural basis of behavior.1) Much of this work has already been published (Murphy et al., 2016). Details provided here are improvements upon the original methods, rather than new findings or techniques.

We now better describe how this cage is an advance over what was previously published. In addition to having more features, including lick detection, task integration, behavioral camera triggering, and automatic weighing, the current system is much better documented and includes a fully illustrated guide to cage construction (Home cage Construction Guide). We feel this guide is essential for anyone planning to construct one of these systems. We’ve also better documented the open-source Python acquisition software and include an SQL database for tracking mouse parameters which can be done using a live cloud-based format. 2) The home cage is now used with longitudinal GCAMP signals from 13 different animals (See Figures 9 and 10). In this group data, we are able to make several conclusions (described in the response to the subsequent comment) regarding the GCAMP dynamics and behavioral outcome in 2 new added figures.

2) Home cage system is described to be optimized for capturing long-term functional changes in cortex, yet the data showed no functional changes after training. Described results did not elucidate benefits of using this system.

In the revision we include functional group analysis of functional imaging data from a total of 13 different mice, see subsection “Decoding of trial outcome”. For simplicity, we analyze group data from a simple go-detection task and were able to create a model that predicts behavioral trial outcome based on brain activity data. This model was used to examine temporal features and regions which exhibited predictive value in the behavioral task. This new analysis is now presented in two additional figures. We have also provided group data on the relationship between body movements and mesoscale brain activity. Here, consistent with other findings, there is a strong correlation between movement and mesoscale activity. Also, consistent with recent data is the finding that individual mice have different strategies for task completion. This work also consistent with findings from the Helmchen lab (Gilad et al., 2018). In total this new group analysis of functional imaging data suggests that our home cage is a well-characterized system that can produce results linking mesoscale networks to behavior.